# Understanding how excess lead iodide precursor improves halide perovskite solar cell performance

Byung-wook Park[1], Nir Kedem[2], Michael Kulbak[2], Do Yoon Lee[1], Woon Seok Yang[1], Nam Joong Jeon[3], Jangwon Seo[3], Geonhwa Kim[4,5], Ki Jeong Kim[4], Tae Joo Shin[6], Gary Hodes[2], David Cahen [2] & Sang Il Seok [1]

The presence of excess lead iodide in halide perovskites has been key for surpassing 20% photon-to-power conversion efficiency. To achieve even higher power conversion efficiencies, it is important to understand the role of remnant lead iodide in these perovskites. To that end, we explored the mechanism facilitating this effect by identifying the impact of excess lead iodide within the perovskite film on charge diffusion length, using electron-beam-induced current measurements, and on film formation properties, from grazing-incidence wide-angle X-ray scattering and high-resolution transmission electron microscopy. Based on our results, we propose that excess lead iodide in the perovskite precursors can reduce the halide vacancy concentration and lead to formation of azimuthal angle-oriented cubic α-perovskite crystals in-between 0° and 90°. We further identify a higher perovskite carrier concentration inside the nanostructured titanium dioxide layer than in the capping layer. These effects are consistent with enhanced lead iodide-rich perovskite solar cell performance and illustrate the role of lead iodide.

[1] Perovtronics Research Center, School of Energy and Chemical Engineering, Ulsan National Institute of Science and Technology (UNIST), 50 UNIST-gil, Eonyang-eup, Ulsan, Ulju-gun 44919, Korea. [2] Department of Materials & Interface, Weizmann Institute of Science, 7610001 Rehovot, Israel. [3] Division of Advanced Materials, Korea Research Institute of Chemical Technology (KRICT), 141 Gajeong-RoYuseong-GuDaejeon 34114, Korea. [4] Beamline Research Division, Pohang Accelerator Laboratory (PAL), Pohang University of Science and Technology (POSTECH), Pohang 37673, Korea. [5] Department of Physics, Gwangju Institute of Science and Technology (GIST), Gwangju 61005, Korea. [6] UNIST Central Research Facilities & School of Natural Science, Ulsan National Institute of Science and Technology (UNIST), 50 UNIST-gil, Eonyang-eup, Ulsan, Ulju-gun 44919, Korea. These authors contributed equally: Byung-wook Park, Nir Kedem, Michael Kulbak. Correspondence and requests for materials should be addressed to G.H. (email: gary.hodes@weizmann.ac.il) or to D.C. (email: david.cahen@weizmann.ac.il) or to S.I.S. (email: seoksi@unist.ac.kr)

Halide perovskite solar cells (PSCs) containing organic cations have already achieved power conversion efficiencies (PCEs) of more than 22% in a very short time[1]. The PCEs of PSCs have been mainly improved by developing methods to deposit highly uniform and dense layers on substrates and to control the $ABX_3$ composition of the halide perovskites (HaP), where X is halide[2]. Compositional control of HaP materials includes varying the ratio of lead iodide ($PbI_2$) and the salt of the "A" cation, methylammonium iodide (MAI) or formaminidium iodide (FAI), used to form the perovskite layers. Bi et al.[3] and our group[4] have independently reported that addition of excess $PbI_2$ to a solution used to prepare an HaP of 15% methylammonium lead bromide ($MAPbBr_3$) mixed with 85% formamidinium lead iodide ($FAPbI_3$), plays an important role in improving the PCEs of PSCs, based on this HaP (($FAPbI_3$)$_{0.85}$($MAPbBr_3$)$_{0.15}$). However, it is not clear how excess $PbI_2$ in the precursor solution, produced by adding $PbI_2$ in excess to the required (MAX or FAX) to $PbI_2$ ratio, and residual $PbI_2$ in the HaP layer contributes to efficiency improvement in these PSCs. At this time, the residual $PbI_2$ location in the HaP and its function remain unclear.

While the effect of excess $PbI_2$ in HaPs has been discussed in several previous investigations[5–7], the details of this phenomenon could not be elucidated, and the hypotheses suggested for the origin of these effects are not comprehensive. For example, it has been suggested that $PbI_2$ can be located at the grain boundaries, the surface of the HaP films as well as at the interfaces between nanostructured titanium dioxide (ns-$TiO_2$) and HaPs[5]. Oesinghaus et al.[8] used grazing-incidence X-ray scattering (GIXS) to investigate how different crystal orientations of HaPs ($CH_3NH_3PbI_3$) are formed with various HaPs deposition methods, such as the popular one/two-step coating. Brenner et al.[9] investigated the conversion of single crystal $PbI_2$ to $CH_3NH_3PbI_3$ with emphasis on structural relations and transformation dynamics and found that while initially the reaction proceeds topotactically, ultimately a destruction/reconstruction process occurs and memory of the original $PbI_2$ morphology is lost. Although this is useful information for understanding the process of film formation and effects of different crystalline orientations, it is not yet possible to correlate crystal orientations with charge transfer mechanisms in terms of solar cell function.

Photoluminescence (PL) spectroscopy and time-correlated single photon counting (TCSPC) have been commonly used to understand charge collection efficiencies and identify charge transfer mechanisms. However, for HaP materials it has been difficult to determine exact values of electron/hole pair diffusion lengths in the actual PSCs, because of the need to develop a model for the charge quenching processes in various HaP layers[10]. Moreover, the applicability of these approaches to HaP layers with mixed compositions should be verified.

Here, we examine key features of the electrical and crystal properties of HaP layers with and without excess $PbI_2$, using electron-beam-induced current (EBIC) measurements, grazing-incidence wide-angle X-ray scattering (GIWAXS), and high-resolution transmission electron microscopy (HR-TEM). Our EBIC results suggest that excess $PbI_2$ results in reduced electronic defects in the bulk film, and GIWAXS shows that the $PbI_2$-rich HaP film is textured and has more-ordered grains than the normal film. Additionally, HR-TEM observations indicate that excess $PbI_2$ surrounds the perovskite grains rather than occuring just at grain boundaries. These features may improve charge transfer and lead to very low J-V hysteresis in a PV cell made with this material, compared with what is the case for films and cells made from a stoichiometric solution precursor.

## Results

**EBIC measurements.** EBIC measurements can probe carrier diffusion lengths, $L_n$ and $L_p$, which is a critical material/device property for solar cells. We should use a special form of EBIC, plan-view, on full devices to obtain $L_n$ and $L_p$, because the thickness of the active absorber in the high efficiency devices is less than the diffusion lengths. As this is not the usual way in which EBIC is used to extract diffusion lengths, we then describe how, in these samples, we can use plan-view EBIC and give the working assumptions for use of plan-view EBIC due to the inability to measure diffusion lengths in cross-sectional EBIC images. A model for charge separation mechanism as well as estimates of carrier diffusion lengths can be deduced from EBIC collection efficiency mapping of a PV device, in this case, fluorine-doped tin oxide (FTO)/dense-$TiO_2$/ns-$TiO_2$/Perovskite/PTAA(polytriarylamine)/Au. Thus, EBIC measurements were carried out for two representative types of HaP samples, namely, [$FAPbI_3$]$_{0.85}$[$MAPbBr_3$]$_{0.15}$ (denoted as control-HaP) and [$FAPbI_3$]$_{0.85}$[$MAPbBr_3$]$_{0.15}$, which was prepared with 7.5 mol% of excess $PbI_2$ (denoted as w/-$PbI_2$-HaP). Excited carrier diffusion lengths can be determined from EBIC measurements only if the carrier motion is not subject to any electric field, but such lengths can also be derived from the drift lengths. Carrier diffusion and drift are correlated by the Einstein relation, $D = \mu kTq^{-1}$, where D is the diffusion length, $\mu$ is the carrier mobility, and k, T, and q are the Boltzmann constant, absolute temperature, and carrier charge, respectively. In a p-i-n device, where an electric field is present throughout the absorber film, an upper limit of the diffusion length can be estimated from plan-view EBIC images. There, EBIC mapping is done at the edge of the Au contacts for holes, and at the FTO edge for electrons (Fig. 1a, b). To make it possible for carriers to be collected, carrier motion must be perpendicular to the built-in field and we can assume, to a first approximation, that the carrier motion is not affected by the field that exists across the layer (i.e., 90° from the direction of measurement in plan view). The resulting estimate of the diffusion length is considered as an upper limit, because of the fast extraction of the complementary carrier type to the selective contact. In this case an uneven concentration of excited holes and electrons in an already low-doped semiconductor results in a low recombination rate. In order to estimate excited carrier diffusion lengths from plan-view EBIC, we assume significant lateral transport in the PTAA is unlikely due to its high resistance compared to that of the HaP film. This is due to a 2 to 3 orders of magnitude higher hole mobility in the HaP than in the PTAA and also the difference in the layer thickness (over 500-nm-thick HaP layer and less than 50 nm for the PTAA)[11–13]. Lateral transport of electrons in the d-$TiO_2$ and the incorporated HaP in the nanoporous $TiO_2$ (HaP@ns-$TiO_2$) may be more efficient than that of holes in the PTAA and contribute to the EBIC signal as the e$^-$-beam scans away from the FTO (over parts from which the FTO was etched). Reduced concentration of one type of carrier due to fast extraction of one of the carrier types along with point excitation would increase the effective apparent diffusion length of the other carrier by lowering the probability of second order recombination. A cleaner estimate of the exact diffusion lengths in the HaP would require deposition of the HaP layer, on glass, with two selective contacts, which do not sandwich the HaP film. Deposition over glass requires adjusting the HaP deposition process and may result in a different electronic quality material. The estimated upper limit represents the potential of the material under optimal conditions and in a real device.

Detailed features of the sample setup for electron and hole diffusion length measurements using the EBIC system are given in Supplementary Fig. 1.

We find that the EBIC signal is high and constant up to the FTO edge and to the point where the Au layer becomes

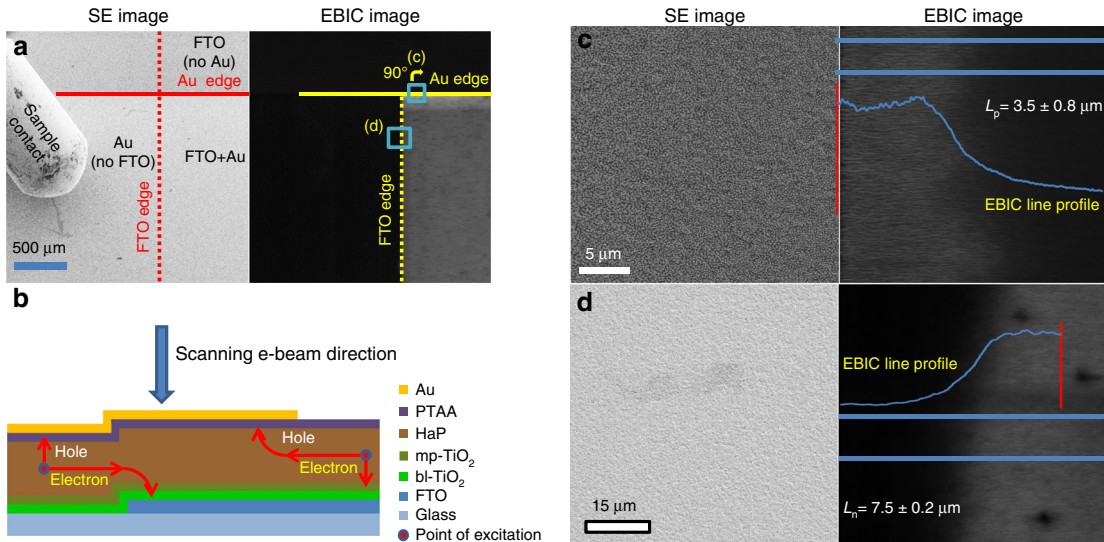

**Fig. 1** Plan-view images of a device, using secondary electron (SE) contrast (left) and EBIC measurements (right). In all images, the TiO₂/perovskite/PTAA (polytriarylamine) layers are present. The changes are only in the presence or absence of the electrodes to the selective contacts, FTO or Au. **a** The edges of the Au pad and FTO are indicated for clarity. The apparent brighter layer close to the Au edge is an artifact of the scan parallel to an interface where charge extraction occurs. The areas that are shown in **c** (turned 90° clockwise) and **d** are indicated as blue squares. **b** Illustration of the device showing the edges of the FTO and Au electrodes (**c**) Expanded SE (left) and EBIC (right) images at the Au pad edge of the device, which is rotated by 90 degrees with respect to **a**. **d** Expanded SE (left) and EBIC (right) images at the FTO edge of the device. Owing to coverage by multiple layers, the FTO edge cannot be resolved in the SE image. Using EBIC contrast, the FTO edge can be located clearly

discontinuous. Beyond these borders, the signal starts to decay, which allows measurement of the electron ($L_n$) and hole ($L_p$) diffusion lengths. We can extract the diffusion length from the EBIC data by fitting the EBIC current to a simple exponential decay,

$$I = I_0 + Ae^{[-X/L_{n/p}]} \qquad (1)$$

where $I$ is the signal intensity, $I_0$ is the intensity at maximal collection efficiency, A is an arbitrary intensity coefficient, $X$ is a spatial coordinate and $L_{n/p}$ is the diffusion length for electrons/holes, as described by Donolato et al.[14]. The 1-D model, on which equation (1) is based, assumes that excited carrier motion under neutral field conditions is limited by the diffusion length. In such a model, where the carrier motion is perpendicular to the line across which carrier separation occurs, the current can be modeled by an exponential decay as described by Equation (1). Such analysis (the fitting process is illustrated in Supplementary Fig. 2) results in values of $L_n = 7.5 \pm 0.2$ μm and $L_p = 3.5 \pm 0.8$ μm, as shown in Fig. 1c, d. The electron diffusion length is longer than the hole diffusion length, as commonly observed in semiconductors with a lighter effective mass of electrons than of holes[15], and as was previously found for MAPbI₃[16]. Notably, no significant differences were found in the $L_p$ and $L_n$ values of the control-HaP and w/-PbI₂-HaP samples (Supplementary Fig. 2). The similar excited carrier diffusion lengths in the two samples are not surprising for two main reasons: first, both sample types show similar short-circuit current (Supplementary Fig. 3, EBIC is done at short-circuit condition). Second, the diffusion lengths are upper limits, i.e., for unipolar transport as the second carrier is collected immediately (the lower limit is the layer thickness). Under operating conditions, closer to $P_{max}$, a more pronounced difference in diffusion lengths may be seen. Bias application during EBIC imaging, in order to simulate $P_{max}$ conditions, is tricky and beyond the scope for this work.

Analysis of beam-induced changes in plan-view EBIC mapping shows degradation of the signal with repetitive exposure. As the

plan-view imaging does not include any pre-processing of the sample, the degradation can be attributed to defect formation due to the electron beam. As previously seen in similar cases, the beam-induced damage does not alter the diffusion length estimation in the first few exposures, but merely lowers the total EBIC signal[17,18]. We note that in earlier EBIC studies there was no need to develop plan-view EBIC, because samples were used with absorber layers that were deliberately fabricated much thicker (2 μm) than what is needed for a PV cell (~0.5 μm)[16]. In such samples the diffusion lengths of both electrons and holes can be measured in a cross-sectional solar cell configuration, which is not possible in the top PV-quality cells, used here.

To understand the defect properties of control-HaP and w/-PbI₂-HaP, we investigated changes in the EBIC signal during repetitive scans of cross-sectional samples (Fig. 2). Beam-induced changes in the EBIC signal are commonly observed for HaP-based devices[17]. As shown in Fig. 2a, the EBIC signal intensity can be divided into two parts, depending on the sample, with the signal strength increasing for the first 1–5 scans, and further scans leading to a decay in the signal strength. An increase in EBIC signal strength with exposure is well known for Si-based devices[19] and has also been found and described for MAPbI₃ devices[17].

Two possible mechanisms may result in the signal peaking after a few exposures: first, initial passivation of (surface) trap states, which leads to suppression of (near) surface recombination, and which initially increases the EBIC signal. A peak is formed if the beam-induced damage also creates damage to the material, which at some point becomes significant and results in measurable signal reduction; second, beam-induced change in the effective doping concentration, which is initially optimized with respect to the first scan, but later forms an excess, which results in an increased recombination rate and, as a result, a lower EBIC signal.

The latter explanation is independent of whether or not a new surface was created as part of sample preparation for EBIC analysis (i.e., sample cleaving). This means that a peak in EBIC signal with repetitive scans should have appeared both in

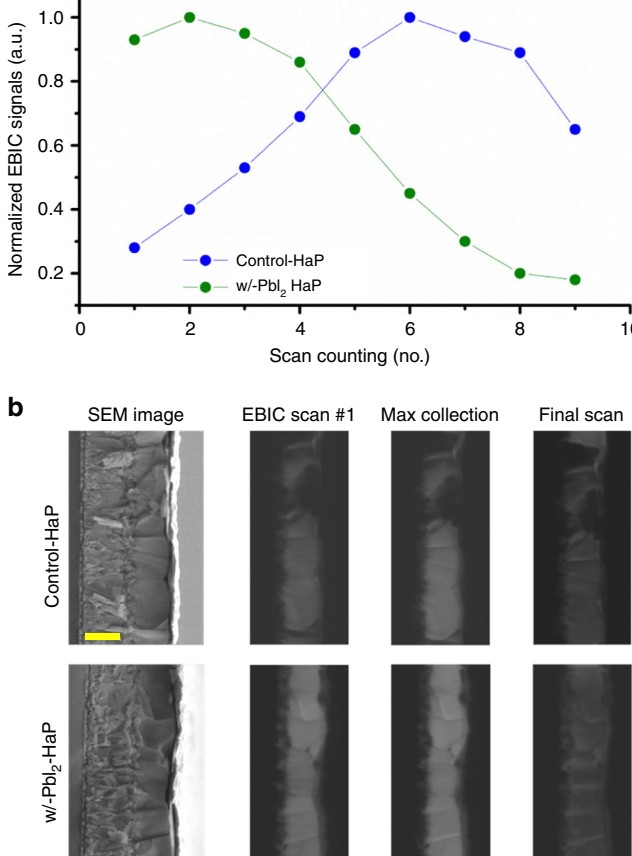

**Fig. 2** The changes in the EBIC signal during repetitive scans of cross-sectional samples. **a** Normalized EBIC signal vs. scan number for control-HaP (blue) and w/-PbI$_2$-HaP (green). The EBIC signal of each sample is normalized to the maximum intensity of that sample. **b** Cross-sectional SEM images of the two samples with EBIC mapping of the cross section after various beam exposure. Scale bars, 500nm

efficiency from that layer is low, and may represent a loss in a real device, due to absorption of light in a poorly photoactive layer. Possible reasons for high recombination rate in the HaP@ns-TiO$_2$ are defects at the HaP/TiO$_2$ interface, high (dark) carrier concentration due to charge transfer from the TiO$_2$ to the HaP across the interface, or defects in the HaP as a result of less optimal crystallization than that of the HaP capping-layer. The relatively low external quantum efficiency at short wavelength, previously reported by us[4], is consistent with such an explanation. Strong optical absorption in the short wavelength range implies essentially complete absorption (in this range) by the HaP in the ns-TiO$_2$ and loss of current. An advantage of having the HaP@ns-TiO$_2$ layer could be to promote charge separation, even before carrier injection into the TiO$_2$ by formation of high electron concentration at the HaP/HaP@ns-TiO$_2$ interface. Such a junction should be selective for electrons and repel holes from the electron extraction interface, which may reduce overall recombination. Further investigation is required to understand the origin of, and thus avoid, the low collection efficiency in the HaP@TiO$_2$ layer.

The differences between the EBIC signals in the presence and absence of excess PbI$_2$ in the deposition solution can be caused by several factors. For example, excess PbI$_2$ in solution may act to shift the crystallization reaction equilibrium towards a material with fewer iodide (I$^-$) vacancies, by maintaining high I$^-$ activity during crystallization and, thus, limiting the vacancy density (assuming that formation energies for interstitials or anti-site defects are too high to make these defects occur in significant concentrations)[20]. This defect formation process can lead to decreased charge recombination lifetimes[1,21,22] due to non-stoichiometry between Pb and I. Alternatively, a thin PbI$_2$ coating at the HaP grain boundaries may somehow passivate defect surface states and prevent recombination (at the surfaces)[23]. We note that the above discussion is limited to bulk grains and the interface between the HaP layer with the selective contacts, and does not include HaP grain boundaries. We also selected only areas of the sample that had a flat morphology. This self-imposed limitation is due to the EBIC signal sensitivity to sample surface morphology.

**GIWAXS investigation.** EBIC-based observations are complemented by comparison between crystal properties of control-HaP and w/-PbI$_2$-HaP and correlating them with the excited carrier collection efficiency and defect concentration. In general, GIWAXS allows a two-dimensional (2D) analysis of crystal orientation, both in and out of the sample plane (see Supplementary Fig. 5, which shows Miller indices for a FAPbI$_3$ thin film). In addition, this analysis technique can provide investigation of the local crystal phase distribution as a function of film depth by varying the Grazing Incident X-ray (GIX) angle (°)[24]. In Fig. 3, GIWAXS 2D patterns of control-HaP and w/-PbI$_2$-HaP are presented at GIX angles of 0.15°, 0.20° and 0.30°, which correspond to film depths of 4, 11–22 and 80–85 nm from the surface of the capping HaP layer, based on calculated penetration depths of the incident X-ray beam (Supplementary Fig. 6).

According to all GIWAXS 2D patterns that we obtained (Fig. 3), both HaP samples contain three types of crystals, viz. trigonal PbI$_2$, hexagonal (δ-), and cubic α-HaP. We note that δ-HaP was not observed in conventional XRD measurements (Supplementary Fig. 7), which suggests that it may be on the surface of the α-HaP crystal or in a very small boundary layer between PbI$_2$ and α-HaP crystals[19]. This hypothesis is consistent with the stronger intensities for δ-HaP [100]$_h$ of w/-PbI$_2$-HaP (Fig. 3f) than in control-HaP (Fig. 3c). In addition, different crystal orientations for α-HaP, [100]c and [200]c, can be observed as preferential GIWAXS peaks for both HaPs. For control-HaP,

plan-view, as well as in cross-section imaging. The appearance of such a peak in the cross-sectioned samples is only relevant for the first explanation, where the HaP has exposed free surface. A lower surface state concentration in w/-PbI$_2$-HaP is indicated by a high initial signal, unaffected by previous beam exposure, and fewer exposures required to reach the maximum signal value. The initial EBIC signal strength at the w/-PbI$_2$-HaP was 70% ± 8% of the highest value (Fig. 2a shows an extreme case of signal strength of ~90% of the maximum), whereas for the control-HaP, 40% ± 16% of the highest signal was observed. 1–3 beam exposures were required to reach the highest signal for w/-PbI$_2$-HaP, and 4–6 for the control-HaP. An example for the evolution of the EBIC signal owing to beam-damage-related signal decay is shown in Fig. 2a, b.

In light of our understanding of the mechanism for beam-induced changes of the EBIC signal, we conclude that the w/-PbI$_2$-HaP is less susceptible to defect formation during cleaving and lab air exposure than the control sample.

A striking observation from the EBIC line profile is the relatively low signal from the HaP@ns-TiO$_2$ layer, only some 8–17% of the signal from the capping HaP film (Supplementary Fig. 4). As an HaP content larger than 25% but considerably less than the content in the HaP overlayer, can clearly be seen in the ns-TiO$_2$ layer using high-resolution Z contrast SEM imaging (Supplementary Fig. 1), the low EBIC signal is attributed to a high recombination rate. This means that the current collection

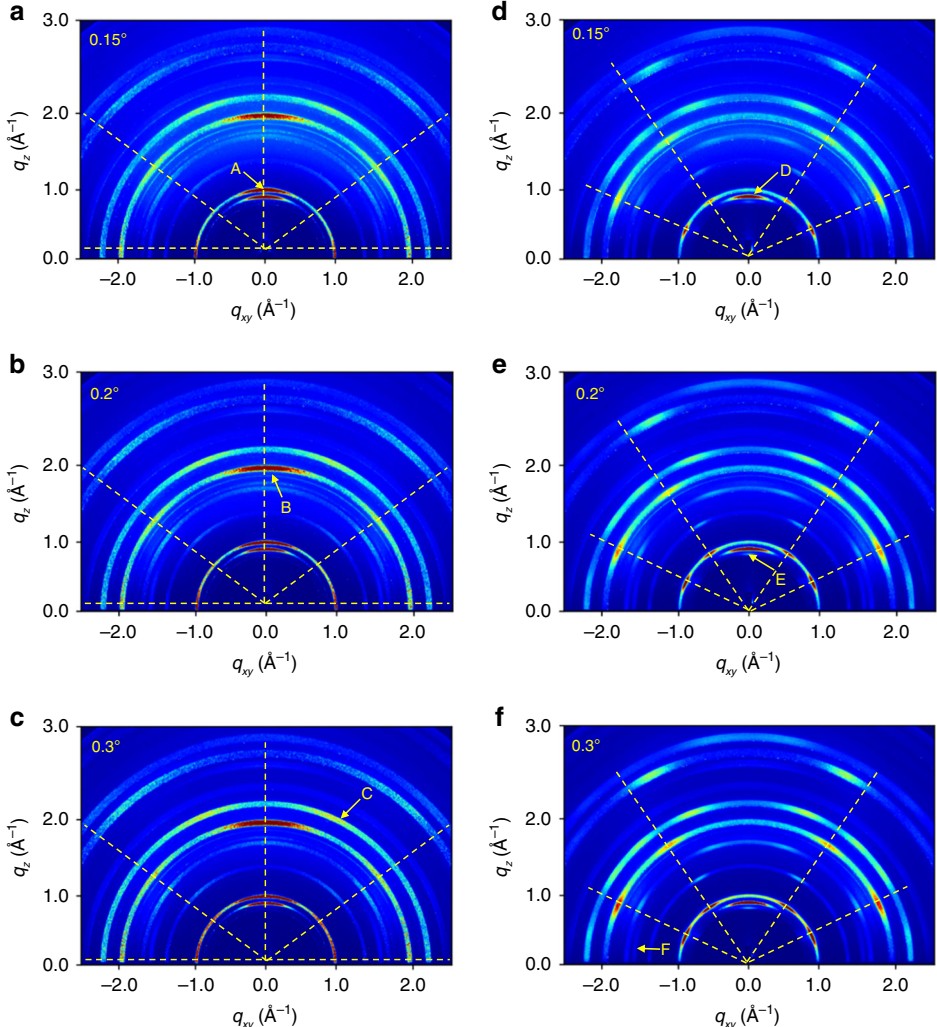

**Fig. 3** GIWAXS 2D patterns for control-HaP (**a**–**c**) and w/-PbI$_2$-HaP (**d**–**f**) at different X-ray grazing-incidence angles such as 0.15°(**a**) and (**d**), 0.20° (**b**) and (**e**), and 0.3° (**c**) and (**f**), measured at 11.6 keV for 10 s. Peaks A, B, C, D, E, and F are assigned to α[100]$_{cubic}$, α[200]$_{cubic}$, α[210]$_{cubic}$, α[100]$_{trigonal}$ of PbI$_2$, δ[100]$_{hexagonal}$, and δ[002]$_{hexagonal}$ in this HaP film, respectively

as shown in Supplementary Fig. 8a, α-HaP[100]$_c$ is mainly observed out-of-plane, and very few crystallites are oriented at 5° (175°) and 40° (140°) of azimuthal angles, where the angle in parentheses represents the symmetry value of the GIWAXS 2D pattern. At the same time, in the case of w/-PbI$_2$-HaP (Supplementary Fig. 8b), α-HaP [100]c is oriented at 24° (156°) and 56° (124°) of azimuthal angles. This α-HaP texturing is illustrated schematically in Supplementary Fig. 8c and d; it may play a role in the overall electronic properties of the film due to alignment of directions of polarization in general, and of ferroelectric domains, where those are present[25,26].

For both types of HaP films the orientation of all α-HaP crystallites is more random deeper into the film. PbI$_2$ remnants in both HaP sample types show similar out-of-plane orientation, but with rather different distributions: broader peaks in azimuthal direction for control-HaP (Supplementary Fig. 8e) and sharper peaks for w/-PbI$_2$-HaP (Supplementary Fig. 8f). As already reported[27,28], the formation of a crystalline HaP thin film involves a process in which precursors, dissolved in a solution, are first solidified in the form of a compound bound to a solvent, and as the solvent is removed crystallization occurs with the help of external thermal energy. HaP crystal transformations from the as-prepared film were monitored from 6 s to 480 s on a hot plate during heating from 30 °C to 150 °C by in-situ GIWAXS, to

reveal when a PbI$_2$ remnant was produced, and when the crystalline HaP was oriented. The as-prepared film was obtained by dripping diethyl ether as an antisolvent onto the substrate during spin-coating of the perovskite precursor solution. At the initial stage after 6 s, it was found in Fig. 9 that α-FAPbI$_3$ [200] and δ-FAPbI$_3$ [100] were formed at an early stage. We see that the phase of the δ-FAPbI$_3$ [100] is significantly reduced and converted to α-FAPbI$_3$ and PbI$_2$ [001] phase at 60 s. No significant changes occur between 60 and 480 s. This implies that an excess PbI$_2$ in precursor solution forms a thin intermediate layer of the FAI deficient-FAI-PbI$_2$-DMSO[29] complex rather than PbI$_2$-DMSO, after spin-coating, followed by the conversion of α-FAPbI$_3$ [100] and PbI$_2$ [001] phase from δ-FAPbI$_3$ [100] with annealing at high temperature. This result also explains why PbI$_2$ is present inside the α-FAPbI$_3$ grains, as will be seen later in the TEM. We now take a closer look at the changes taking place. After annealing for 6 s, a non-perovskite phase derived from yellow δ-FAPbI$_3$ was observed at $q = 0.84$ Å$^{-1}$. Interestingly, this phase showed two distinct orientations at azimuth angles of 30° (150°), and 90°. After further heating for 60 s, peaks at $q = 0.84$ Å$^{-1}$ began to disappear, whereas another diffraction peak at $q = 1.0$ Å$^{-1}$ originating from black α-FAPbI$_3$ appeared with two specific orientations at azimuth angles of 25° (155°) and 55° (125°). The in-situ GIWAXS shows that the phase transition during

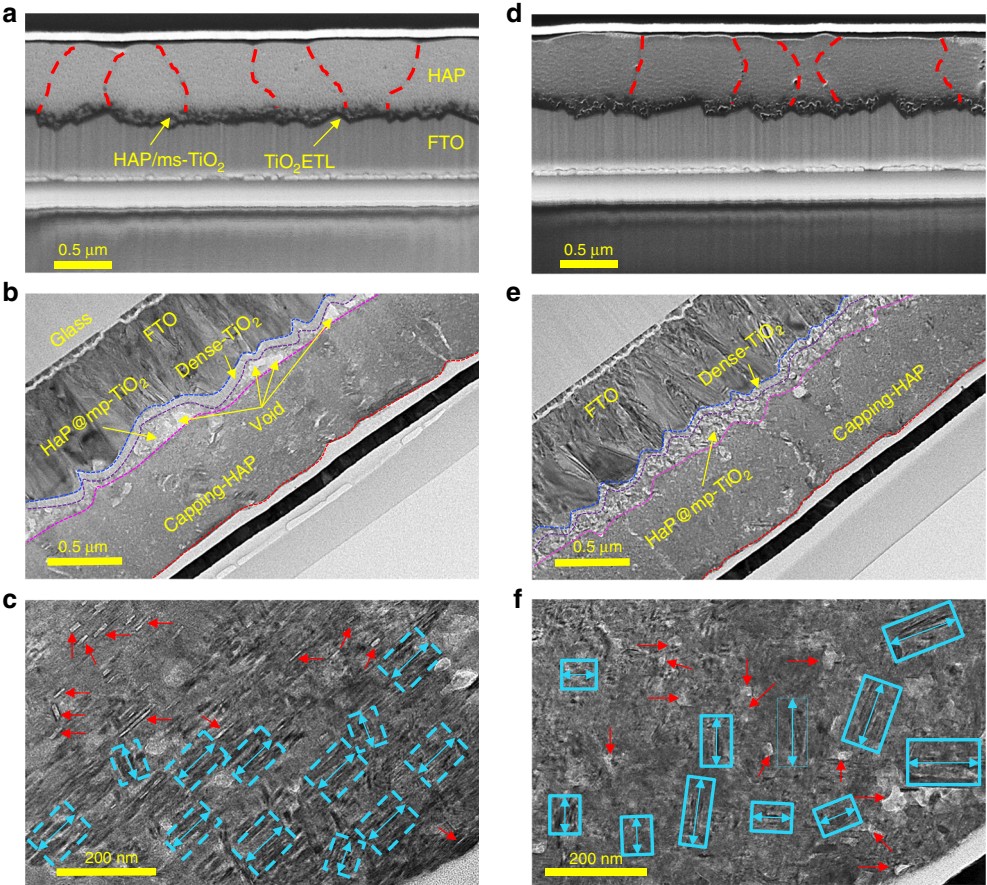

**Fig. 4** FE-SEM and HR-TEM cross-sectional images (**a** control-HaP, **d** w/-PbI$_2$-HaP) and HR-TEM cross-sectional images (**b** control-HaP, **e** w/-PbI$_2$-HaP). HR-TEM images **c** and **f** show magnifications of images **b** and **e** respectively, of the HaP layer area. Red arrows indicate unreacted HaP with PbI$_2$ residues (or clusters) and blue arrows in rectangles (dashed or solid) correspond to the direction of lattice planes

crystallization and growth does not significantly affect the orientation of the perovskite crystallites. This preferred orientation of the perovskite crystallite is presumed to be determined from the intermediate complex due to differences in the precursor solution, but further study is required. The importance for this study is that no peak of PbI$_2$ [001]$_t$ is seen in Supplementary Fig. 9. Thus, we conclude that remnant PbI$_2$ does not appear initially, but is slowly formed during annealing as reported earlier for MAPbI$_3$[23].

**HR-TEM observation**. To follow up on indications from GIWAXS experiments, we carried out cross-section SEM and HR-TEM analyses to study the structures and distribution of the crystallites. SEM analysis suggests that there may be a slightly larger domain size for the w/-PbI$_2$-HaP sample than for the control-HaP sample (compare Fig. 4a, d), as previously reported[4]. Such a difference may indicate a change in formation energy when a small excess of PbI$_2$ is added to the HaP precursor solution. Higher magnification TEM images (Fig. 4b, c, e, f), reveal the microstructures of the domains, with different lattice plane directions (blue rectangles and red arrows in Fig. 4c, f), observed for each sample (Supplementary Figs. 10 and 11). PbI$_2$ inclusions (indicated by red arrows) can be distinguished from the HaP matrix by their brighter color. In the case of control-HaP, a significant amount of PbI$_2$ particles appears to be arranged out-of-plane, relative to the lattice plane direction, and we assume that this observation indicates the formation of PbI$_2$ particles within the HaP grains, as shown in Supplementary Fig. 10. At the

same time, small, elongated particles of PbI$_2$ are observed for w/-PbI$_2$-HaP. When organic cations diffuse during the annealing process, PbI$_2$-like clusters that are assigned to 'A' in Supplementary Fig. 11b can react with the organic cations to form δ-HaP at the interface (assigned to "B" in Supplementary Fig. 11b).

HaP film texturing, i.e., the orientation of crystallites within the film with respect to each other, may play a role in the performance of HaP-based devices. This texturing will affect the alignment of the ferroelectric domains, which can be correlated with desired pathways for charge extraction (transfer) and is presumed to be one of the factors leading to a high $V_{oc}$ seen in w/-PbI$_2$-HaP. Although the texturing is not accompanied by changes in lateral charge diffusion lengths as obtained from the EBIC (Fig. 1), we showed earlier that optimization of excess PbI$_2$ in the deposition solution can improve fill factor (FF) and short-circuit current density ($J_{SC}$) (also, external quantum efficiency), and increase shunt resistance ($R_{sh}$). In particular, hysteresis in the $J$-$V$ scan was suppressed (Supplementary Fig. 3)[4,5].

A, likely critical, effect of adding PbI$_2$ to the HaP precursor solution for film formation is that for w/-PbI$_2$-HaP films the ns-TiO$_2$ is nearly completely filled by HaP, while voids are seen in the control-HaP films, as shown in Fig. 4b. Indeed, we observe in Supplementary Fig. 12 that the addition of PbI$_2$ to the HaP precursor solution results in a smaller iodoplumbate complex size. This is illustrated also in a high magnification HR-TEM image for a representative small volume of the w/-PbI$_2$-HaP film (Supplementary Fig. 11c), which suggests that the HaP coats the surface of ns-TiO$_2$. Thus, it seems that a w/-PbI$_2$-HaP precursor solution yields films that fill the ns-TiO$_2$ better than what is the

case for films made with the control solution. These observations can explain the lower stability of the EBIC signal of HaP@ns-TiO$_2$ films, made without excess PbI$_2$ (Fig. 2).

## Discussion

We have elucidated the influence of PbI$_2$ added to HaP precursors before HaP film formation. Comparing two types of samples, control-HaP and w/-PbI$_2$-HaP thin films, we find the same electron and hole diffusion lengths for both types of samples from EBIC measurements ($L_n = 7.5 \pm 0.2$ μm and $L_p = 3.5 \pm 0.8$ μm,). However, the EBIC signal decay upon repetitive scans is stronger for the control-HaP films, suggesting a lower defect concentration in w/-PbI$_2$-HaP than in control-HaP. GIWAXS 2D patterns and HR-TEM images revealed differences of α-HaP crystal orientation between the two types of HaP samples. In light of these data, we conclude that use of excess PbI$_2$ in HaP precursor leads to α-HaP crystals with lower defect concentrations and more oriented crystal lattice directions from the surface into the crystallites. Those differences may well be (part of) the reasons for enhanced device performance[4,5] as well as lower J-V hysteresis of PSCs made with these films. While it is not yet possible to give a full description for the mechanism that governs α-HaP crystal orientations in the initial part of the film coating process, we foresee that further clarifications of the excess PbI$_2$ effects will allow to formulate guidelines for development of higher PCEs devices as well as improved stability of the cells.

## Methods

**Fabrication of solar cell device**. A 60-nm-thick dense blocking layer of TiO$_2$ (bl-TiO$_2$) was deposited onto an F-doped SnO$_2$ (FTO, Pilkington, TEC8) substrate by spray pyrolysis, using a $20 \times 10^{-3}$ M titanium diisopropoxide bis(acetylacetonate) solution (Aldrich) at 500 °C, to prevent direct contact between FTO and the hole-conducting layer. 80-nm-thick mesoporous TiO$_2$ (home-made TiO$_2$ nanoparticles: average particle size is around 50 nm, crystalline phase is anatase) films were spin-coated onto the bl-TiO$_2$/FTO substrate and were calcined at 500 °C for 1 h in air to remove the organic portion. To fabricate efficient perovskite cells based on the (FAPbI$_3$)$_{0.85}$(MAPbBr$_3$)$_{0.15}$:PbI$_2$, 1.05 M solutions with the ratio of 0.85(NH$_2$CH = NH$_2$(=FA)PbI$_3$), 0.15(CH$_3$NH$_3$(=MA)PbBr$_3$) and MACl (in case of excess PbI$_2$ 7.5 mol% PbI$_2$ powder was added to this precursor solution) in N-N-dimethyl-formamide(DMF) and dimethylsulfoxide(DMSO) = (6:1 v/v) were then coated onto the nanostructured(ns)-TiO$_2$/bl-TiO$_2$/FTO substrate, heated to 50 °C, by two consecutive spin-coating steps, at 1000 and 5000 rpm for 5 and 10 s, respectively. During the second spin-coating step, 1 mL ethyl ether was poured onto the substrate after 5 s. A polytriarylamine (PTAA) (EM index Co. LTD., Mn is 17500 g mol$^{-1}$)/toluene (10 mg mL$^{-1}$) solution with 7.5 μL Li-bis(trifl uor-omethanesulfonyl) imide (Li-TFSI)/acetonitrile (170 mg mL$^{-1}$) and 4 μL 4-tert-butylpyridine (TBP) added, was spin-coated on the perovskite layer/ns-TiO$_2$/bl-TiO$_2$/FTO substrate at 3000 rpm for 30 s. Finally, 80 nm of Au was deposited as electrical contact by thermal evaporation. The active area of this electrode was fixed at 0.16 cm$^2$.

**EBIC measurement**. Control—and w/-PbI$_2$-HaP—samples were prepared according to a previously reported method[4]. EBIC imaging was conducted with a LEO Supra scanning electron microscope and a Stanford Research Instruments pre-amplifier (SR570) to amplify the current and convert it to voltage feed for the instrument computer. Plan-view imaging was performed at 5 and 10 kV beam energies. Cross-sectional imaging was performed at 3 kV beam energy, after the sample was mechanically cleaved and transferred to the vacuum chamber within less than 2 minutes. In order to cleave the sample, a small, straight cut is made on the glass (uncovered) side of the substrate across the substrate using a glass-cutting pen. Then, compressive pressure is applied at one of the contacts (FTO or the etched glass/gold) to fracture the device.

In all images, the aperture diameter was 7.5 μm, the beam current was 3–4 pA, and the working distance was 5 to 7 mm. Images were collected in pixel average mode. Under these conditions, the interaction volume limits the resolution to around 20 nm.

Statistics for all EBIC data were obtained from two sets of w/PbI$_2$-HaP and control samples, prepared independently. Each set was composed of two samples from each type. For diffusion length estimation each sample was measured five times, at different locations. At each location the average of 25 EBIC lines was used to arrive at a single calculated value. The five values from each sample were compared between all samples of the same type, i.e. 20 values per sample type, to form an average value and standard deviation.

For the total intensity decay analysis, an averaged EBIC signal was taken from a 2D square map. The map area was 5 consecutive pixel lines across the HaP layer. The intensity decay analysis was performed by repetitive scans of the same area. Quick scans (up to 3 seconds) are done to center a region of interest without leaving significant damage. Then, every image scan takes 2.8–3.0 minutes while re-taking the images is done in less than 3 seconds, in which the current image is recorded and immediately followed by a re-scan of the area. We did not observe beam damage dependence on time between scans. Next, the intensity at each location was normalized according to the maximum achieved intensity for that area. An averaged initial intensity value was obtained by a similar averaging process as described above. The fitting of the data results in an $R^2$ value of 0.99 and 0.94 for $L_n$ and $L_p$.

Care was taken to account for beam-induced changes in the signal, which can affect the measured properties[15]. For each diffusion length measurement (planar view, which does not involve sample cleaving), repetitive scans were performed to study the signal decay and compare it to cross-sectional decay as described in the text. Since our setup does not provide absolute EBIC current, we consider relative intensity only.

**GIWAXS investigation**. GIWAXS measurements were conducted on the PLS-II 6D UNIST- PAL beamline of the Pohang Accelerator Laboratory in the Republic of Korea. The X-rays from the bending magnet are monochromatized to $\lambda = 0.62382$ Å, using a double crystal monochromator and focused both horizontally and vertically (120 (V) × 150 (H) μm$^2$ in FWHM at sample position) using the sagittal Si(111) crystal and toroidal mirror. Incidence angles of 0.15°, 0.2°, and 0.3° were used in this study. The vacuum GIWAXS sample chamber is equipped with a 5-axis motorized stage for fine sample alignment. The incidence angle of the X-ray beam was set to 0.15°, which is close to the critical angle for HaP. Two-dimensional GIWAXS 2 dimension (2D) patterns were recorded with a 2D CCD detector (Rayonix MX 225-HS, USA). Diffraction angles were calibrated by a pre-calibrated sucrose (Monoclinic, P2$_1$, $a = 10.8631$ Å, $b = 8.7044$ Å, $c = 7.7624$ Å, b = 102.938°)[24] and the sample-to-detector distance was 246.4 mm. The samples were exposed to X-ray beam energy of 11.6 keV for only 10 s to prevent damage. GIWAXS 2D images were converted to one-dimensional patterns using IGOR Pro.

**HR-TEM observation**. To investigate local crystal orientations, most samples were prepared using a Helios NanoLab 450 model focused ion beam, FIB (FEI) and FE-SEM. The FIB milling/polishing process ensured that the samples had a smooth surface. Furthermore, the FIB-based milling process allowed the sample to be thinned to 100 nm, as required for HR-TEM characterization. HR-TEM images were collected using a JEOL JEM-2100F instrument.

**Characterization of solar cells**. The J-V curves were measured with a source meter (Keithley 2420) using a solar simulator (Newport, Oriel Class AAA, 94043 A) at 100 mA·cm$^{-2}$ illumination (AM 1.5 G) and a calibrated Si reference cell (Newport, Model 91150 V) certificated by NREL. The calibrated 1 sun illumination is automatically maintained by exposure control (Newport, Model 68951). The J-V curves were measured in reverse scan modes. The step voltage and delay time were fixed at 10 mV and 40 ms, respectively. The delay time is used at each voltage step before measuring the current. The J-V curves of each device were measured by masking the active area with a metal mask (area of 0.096 cm$^2$). External quantum efficiencies for each HaP solar cell were measured, using an IQE- 200B system (Newport, Oriel).

**Data availability**. The data that support the findings of this study are available from the corresponding author upon request.

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

## Acknowledgements

This work was supported by the Global Frontier R&D Program for Multiscale Energy System (NRF-2011-0031565), the Climate Change Program (NRF-2015M1A2A2056542), and the Basic Science Research Program (NRF-2017R1D1A1B03029832) through the National Research Foundation of Korea (NRF) funded by the Ministry of Science and ICT (MSIT). This work was also financially supported by a brand project (1.180043.01) of UNIST and Ulsan city (1.180030.01). The Weizmann Institute authors thank Dana and Yossie Hollander via the Weizmann Institute's Sustainability and Energy Research Initiative and the Israel Ministry of Science for partial support. D.C. held the Sylvia and Rowland Schaefer Chair in Energy Research. We thank Gyeong-Ae Lee and Sun Yi Lee for partial supporting to conduct HR-TEM and FIB sampling of HaP samples in UCRF.

## Author contributions

B.-W.P., N.K., G.H., D.C. and S.I.S. conceived the experiments and have done data analysis and interpretation of data. B.-W.P., D.Y.L., W.S.Y. and N.J.J. carried out the synthesis of materials for perovskites and the fabrication of devices and device performance measurements. Characterization of SEM, HR-TEM and GIWAXS were performed by B.-W.P., J.S., G.K., K.J.K. and T.J.S. EBIC was measured by N.K. and M.K. The manuscript was written and revised by B.-W.P., N.K., M.K., G.H., D.C. and S.I.S. The project was planned, directed, and supervised by D.C. and S.I.S. All authors discussed the results and commented on the manuscript.

## Additional information

**Competing interests:** The Authors declare no competing interests.

