## [Peer Review File · Nature Communications]

Reviewers' Comments:

Reviewer #1:

None

Reviewer #2:

Remarks to the Author:

In this article, Park et al. present an interesting study on investigating the origin of solar cell performance improvement due to excess lead iodide in halide perovskite precursor solutions. They employed electron-beam-induced current (EBIC), grazing-incidence wide-angle X-ray scattering (GIWAXS), and high-resolution transmission electron microscopy (HRTEM) to explore the mechanism associated with excess lead iodide. Based on their results, they concluded that excess PbI₂ existed within the perovskite grains can reduce iodine vacancy concentration and lead to the formation of textured cubic phase perovskite crystals and larger crystallite size. The manuscript is well prepared and content-rich. I recommend publication after some minor revisions.

Detailed comments are provided as below:

[1] Figure 1a and b are redundant, and one of them should be removed.

[2] It is unclear which sample (control or w-PbI₂) were measured in Figure 1c and d. Although the authors claimed that no significant difference was found in the diffusion lengths of the control and w-PbI₂ samples, experimental data of both samples should be provided. If multiple samples/regions were measured for each type of devices, the statistics of L_n and L_p should be compared and discussed. Additionally, Figure 1c and d could be labeled clearly to show the perovskite/contact regions.

[3] On page 6, the authors mainly discussed the EBIC result with an emphasis on the beam-damage-related decay of two different samples (Figure 2). They attributed the differences in EBIC signals to the formation of iodide vacancy, which limited the charge recombination lifetime. The conclusion sounds reasonable, but why the diffusion lengths derived from EBIC showed no significant difference in these samples? Figure 2a shows the comparison of normalized EBIC intensities, but how about the absolute intensities? Are there any differences at grain boundaries and grain internals of the control and w-PbI₂ samples? The authors should have more discussion on the EBIC results to show the contrast between these samples.

[4] Figure 2b needs a scale bar to show the size of the SEM/EBIC images.

[5] Page 6, Line 152. The authors claimed that junction of perovskite/mp-TiO₂ is an n/n⁺ type heterojunction. Is there any experimental evidence that perovskite is n-type? What is the estimated carrier concentration for the perovskite layer? Are there any differences for the control and w-PbI₂ perovskite?

[6] It would be more readable if the major GIWAXS features in Figure 3 that are discussed in the manuscript can be labeled.

[7] Page 7, Line 168-169. The absence of δ -HaP in the conventional XRD could be due to the formation of δ -HaP on the surface rather than the grain boundaries between PbI₂ and a-HaP.

Reviewer #3:

Remarks to the Author:

Park et al report how the excess of PbI₂ in the perovskite precursors improves the photovoltaic performance of perovskite solar cells. EBIC measurements were applied to estimate the carrier diffusion length and reveal local electrical properties in the devices. The authors also conducted GIWAXS and TEM characterizations to understand how the local crystallite orientation and microscopic morphology correlate with the local electrical properties as revealed by EBIC. The authors found that the excess of PbI₂, which resides in the perovskite grains rather than at grain boundaries, results in low electronic defect density in perovskite films. The excess of PbI₂ also improves the orientation of perovskite crystallites, resulting in better charge collection and suppressed J-V hysteresis. The origin of the excess of PbI₂ in determining solar cell performance

has been reported by multiple groups (Nano Lett. 14, 4158-4163(2014); J. Am. Chem. Soc. 138, 10331–10343(2016); Adv. Energy Mater. 6, 1502206(2016); APL Mater. 2, 091101(2014)). The point-of-views as presented by authors in this work, are not quite convincing to me at the current stage, and not supported well by sufficient experimental evidence. Generally speaking, this work is lack of novelty as compared to the reported similar works as mentioned above. Therefore, this paper should be rejected. My detailed comments are listed below.

1. The use of EBIC to determine local electrical properties is tricky (especially for unstable hybrid perovskites) due to the electron beam induced damage during measurements (These hybrid perovskites are vulnerable to electron beam irradiation), and significant surface recombinations as a result of damage during the sample preparation (actually, the authors did not mention how to prepare the cross-section samples in the main text or experimental section).
2. The variation of the EBIC signal during scans (as shown in Figure 2) is also questionable for the measurements of unstable perovskite materials. This can be reflected by the decrease in EBIC signal after multiple scans, which is probably due to the beam-induced damage during imaging.
3. For the caption of Figure 2, these sentences may be placed in the wrong place: "Here, the position of guest molecules is unconfirmed. Finally, (III) the intermediate phase film is converted to perovskite phase with corner-sharing octahedrons via extraction of DMSO guest molecules by annealing process. The perovskite film is extremely uniform and flat because of solid-state conversion from the uniform and flat intermediate phase film."
4. To justify the reliability of the carrier diffusion length, the authors should present the detailed fitting method and mechanism for the estimation of carrier diffusion length ($L_n = 7.5 \pm 0.2 \text{ um}$ and $L_p = 3.5 \pm 0.8 \text{ um}$)
5. As shown in the EBIC images of Figure 1a-b, it appears that two layers can be seen in Au side (close to the Au edge, clearly different EBIC contrast in the top as compared to the other part), What is that layer ? In addition, several dark spots are observed in the EBIC images (e.g. Figure 1d), what are those dark spots?
6. For the EBIC images, in order to help to understand the EBIC images, the authors are suggested to provide the intensity scale of EBIC signals.
7. The diffraction peaks should be indexed for all the GIWAXS patterns (Figure 3).
8. The authors claim that "both HaP samples contain three types of crystals, viz. trigonal PbI₂, hexagonal (δ -), and cubic (α -) HaP". The diffraction peak for trigonal PbI₂, hexagonal (δ -), and cubic (α -) HaP, should be identified respectively in the GIWAXS pattern (Figure 3).
9. The authors also claim that "We note that in conventional XRD measurements δ -HaP was not observed (Figure S6), which suggests that it may form as very tiny boundary layer between PbI₂ and α -HaP crystals.²⁰ This hypothesis is consistent with the stronger intensities for δ -HaP [100]h in w/-PbI₂-HaP (Figure 3f) than in control-HaP (Figure 3c)." Which peak is for δ -HaP [100]h ?
10. The key information of how the excess of PbI₂ induces the preferred orientation of the perovskite crystallites is still not clear in this work. And it is not clear how to correlate the orientation of the crystallites with the EBIC in the local region.
11. The authors claim that the excess PbI₂ resides in the perovskite grains rather than at grain boundaries based on the TEM studies. However, this claim should be revisited. To my understanding, the PbI₂ inclusions are in the perovskite matrix, in the other words, the PbI₂ inclusions are surrounded by the perovskite crystals (or grains). These independent small grains may form a large grain with grain boundaries. Thus, the PbI₂ may be considered as residing in the grain boundaries.
12. The solar cell device fabrication details should be provided in the experimental section. In addition, the characterization method for J-V curves (photovoltaic performance) should be provided. Since the short-circuit current density is so high, the authors should provide the EQE curves for two devices.

Reviewer #4:

Remarks to the Author:

Review report

How can lead iodide excess in halide perovskite precursors result in improved solar cell performance?

General statement

The authors target a relevant topic (PbI₂ in mixed perovskites), which has been shown to be of importance for high efficiency perovskite solar cells, and is something which due to measurement difficulties is something that still not is properly understood. They use some advanced characterisation techniques to add more data that can be useful to further understand the importance of PbI₂ in perovskite films.

The topic, the scope, and the techniques used makes it of potential interest for nature communications and its readers. I think the work in principle is publishable. There is, however, also problems with the paper and I think it requires a mayor revision before it potentially could be published.

The paper is not very well written, the descriptions are a bit confusing, and the results, their implications, and how they are extracted requires better descriptions. Below follows some comments I have that I think should be addressed.

Comments

The EBIC measurements.

I do not have personal experience with EBIC, but most of the intended readers probably do not have that either. I think the description of the experiments are rather confusing.

I would like a better description of things like:

- EBIC can probe this by using this effect
- With this sample geometry we therefor expect to see this and higher/lower signals for this/thats regions
- If we do, we can based on this reasoning extract this information in this way
- In our samples we see this
- Because we see this, we extract this information in this way
- This information then implies those interesting things

The logic and the flow of the EBIC section must be written in a clearer, and more logical coherent way.

In figure 1. Is (b) exactly the same as (a) but with annotation?

What is the geometry of the sample? Is this a top view or a cross section?

You have indicated the Au edge and the FTO edge. Pleas also indicate which part that is Au and which part that is FTO.

Is the bottom right Au, and is the bottom left FTO? If that is so, how is the sample prepared? Is the gold lying on of the hole conductor? If so, how did you manage to not deposit the hole conductor on top of the exposed perovskite?

If the bottom left is FTO, how did you remove both the perovskite and the TiO₂ under layer?

Figure S.2 give some guidance, but more descriptions is needed to clearly show the reader what was done. Figure S.2, or some extended version of it, should definitely be a part of the main manuscript, and could be included in figure 1.

Line 74 and 80. You first claim that EBIC require information of the number of defects and then you claim that EBIC indicate that there is less defects. That may be true but logically those two statements does not match when they are presented together.

Line 95. If the carrier motion is parallel to the built in field, is it then not by definition effected by

the electric field? Again, I think this would be clearer with a better description of sample geometry, and the intention behind it.

Line 104. Extraction of L_n and L_p . That is a key result and must be described better. You do not show the fit (as you say you do in figure 1). Describe some theory to why it is reasonable to believe that the exponential fit would give those values. This deserved to get some more attention. If you have found any other estimates of the diffusion lengths in the literature, please relate to those.

GIWAXS

Line 77-78. Textured in an azimuthal angle orientation in-between 0-90°. To me, this says absolutely nothing at all.

If a cubic phase is rotated 0-90° that would cover every possible orientation and there would thus not be an alignment?

Does not azimuthal orientation require a fixed point to be referred to make sense?

Why use spherical coordinates to describe the orientation of crystals in a thin film?

Have you considered using Cartesian coordinates instead? I think that would be clearer.

When you talk about orientation of crystals, for it to be interesting you must relate the orientation of the crystals towards the substrate. An illustration would greatly help here and I suggest an additional figure.

Concerning the in-situ GIWAXS measurements

Interesting approach. Hopefully more can be done with this in alter studies (increasing number of temperatures to measure, change the heating ramp, comparing different compositions)

I think it is unclear which composition that is measured. With or without excess PbI_2

TEM

Half of all the TEM images has terrible quality. That is probably a result of problem while converting to pdf during file uploading, but is something that must be fixed.

The domain indicating crystal orientation in the TEM images are about an order of magnitude smaller than the crystal grains as they appear in the SEM images. Can you comment on this?

Perovskites are known to be fragile and delicate materials, and FIB is a rather brutal technique. It would be reasonable to have some discussion concerning the risk for sample damage, and which reasons there are to believe that the prepared samples are representative for the real film.

Other

Line 40. Also $MABr$, $FABr$

Line 43-44 (and 22 and 85). The mixed perovskite is not a mixture of two perovskite phases as the used notations indicate. Data show that it is a one-phase compound. It is thus not $[FAPbI_3]_x[MAPbBr_3]_{1-x}$ but $F_{Ax}MA_{1-x}Pb_{Br}I_{3-y}$

Synthesis

There is a reference to a synthesis procedure. I do not think that is good enough. Digital space is cheap, and how the materials were synthesised, and how samples were prepared, are such

crucial aspects of the work that they should be described in some detail in the manuscript (even if they are almost identical to previous procedures)

Reviewer #2

In this article, Park et al. present an interesting study on investigating the origin of solar cell performance improvement due to excess lead iodide in halide perovskite precursor solutions. They employed electron-beam-induced current (EBIC), grazing-incidence wide-angle X-ray scattering (GIWAXS), and high-resolution transmission electron microscopy (HRTEM) to explore the mechanism associated with excess lead iodide. Based on their results, they concluded that excess PbI₂ existed within the perovskite grains can reduce iodine vacancy concentration and lead to the formation of textured cubic phase perovskite crystals and larger crystallite size. The manuscript is well prepared and content-rich. I recommend publication after some minor revisions.

Detailed comments are provided as below:

[1] Figure 1a and b are redundant, and one of them should be removed.

Our reply: We thank the reviewer for this input, Figure 1a was removed and Figure 1b takes its place and was remade to be clearer.

[2] It is unclear which sample (control or w-PbI₂) were measured in Figure 1c and d. Although the authors claimed that no significant difference was found in the diffusion lengths of the control and w-PbI₂ samples, experimental data of both samples should be provided. If multiple samples/regions were measured for each type of devices, the statistics of L_n and L_p should be compared and discussed → Please see next reply. Additionally, Figure 1c and d could be labeled clearly to show the perovskite/contact regions.

Our reply: We thank this reviewer for helpful comments. We have altered the numbering and text in Figure 1 to make the figure clearer.

Original:

Figure 1. a, Plan-view images of a device using secondary electron (SE) contrast (left) and electron-beam-induced current (EBIC) measurements (right). **b**, The edges of the Au pad and FTO are indicated for clarity. **c**, Expanded SE (left) and EBIC (right) images at the Au pad edge of the device. The Au pad edge is difficult to resolve in the SE image, but it is clearly exposed by EBIC contrast. The EBIC line profile, averaged from all the pixel lines between the blue lines, is overlaid on the image. **d**, Expanded SE (left) and EBIC (right) images at the FTO edge of the device. Owing to coverage by multiple layers, the FTO edge cannot be resolved in the SE image. Using EBIC contrast, the FTO edge can be located clearly. The EBIC line profile, averaged from all the profiles between the blue lines, is overlaid on the image. To obtain carrier diffusion lengths from these profiles, three positions over two samples were averaged.

Revised:

Figure 1. Plan-view images of a device, using secondary electron (SE) contrast (left) and EBIC measurements (right). In all images, the TiO₂/perovskite/PTAA (poly-triarylamine) layers are present. The changes are only in the presence or absence of the electrodes to the selective contacts, FTO or Au. **a**, The edges of the Au pad and FTO are indicated for clarity. The apparent brighter layer close to the Au edge is an artifact of the scan parallel to an interface where charge extraction occurs. The areas that are shown in (b) [turned 90° clockwise] and (c) are indicated as blue squares. **b**, Expanded SE (left) and EBIC (right) images at the Au pad edge of the device, which is rotated by 90 degrees with respect to (a). The Au pad edge is difficult to resolve in the SE image, but it is clearly exposed by EBIC contrast. The yellow EBIC line profile, averaged from all the pixel lines between the blue lines, is overlaid on the image. The red line marks the area in the image where the EBIC line profile starts from and collection efficiency is maximal. The derived diffusion length for holes from the EBIC line profile is $3.5 \pm 0.8 \mu\text{m}$. **c**, Expanded SE (left) and EBIC (right) images at the FTO edge of the device. Owing to coverage by multiple layers, the FTO edge cannot be resolved in the SE image. Using EBIC contrast, the FTO edge can be located clearly. The EBIC line profile, averaged from all the profiles between the blue lines, is overlaid on the image. The derived diffusion length for electrons from the EBIC line profile is $7.5 \pm 0.2 \mu\text{m}$. The apparent dark spots are probably due to discontinuous HaP film at the edges of the FTO. At the bottom left, paths of electron beam-generated charge carriers are schematically illustrated. When the HaP is excited in an area far away from one of the electrodes, the respective carrier has to diffuse laterally to be collected (red arrows; shown as occurring through the HaP, but may also occur through relevant (hole or electron) charge transport layer). The electron beam scans from above and has to penetrate the Au and PTAA layers.

Concerning the remark:

Although the authors claimed that no significant difference was found in the diffusion lengths of the control and w-PbI₂ samples, experimental data of both samples should be provided. If multiple samples/regions were measured for each type of devices, the statistics of L_n and L_p should be compared and discussed.

Our reply: We added to the experimental (page 14) the following lines:

Statistics for all EBIC data were obtained from two sets of w/PbI₂-HaP and control samples, prepared independently. Each set was composed of two samples from each type. For diffusion length estimation each sample was measured five times, at different locations. At each location the average of 25 EBIC lines was used to arrive at a single calculated value. The five values from each sample were compared between all samples of the same type, i.e. 20 values per sample type, to form an average value and standard deviation. For the total intensity decay analysis, an averaged EBIC signal was taken from a 2D square map. The map area was 5 consecutive pixel lines across the HaP layer. The intensity decay analysis was performed by repetitive scans of the same area. Next, the intensity at each location was normalized, according to the maximum achieved intensity

for that area. An averaged initial intensity value was obtained by a similar averaging process as described above.

Furthermore, we added the following lines on pages 6-7 in order to prevent any confusion regarding the results of L_n and L_p in Figure 1b and c:

The similar excited carrier diffusion lengths in the two samples are not surprising for two main reasons:

1. Both sample types show similar short circuit current (EBIC is done at short-circuit condition, figure S2)
2. The diffusion lengths are upper limits, i.e., for unipolar transport as the second carrier is collected immediately (the lower limit is the layer thickness). Under operating conditions, closer to P_{\max} , a more pronounced difference in diffusion lengths may be seen. Bias application during EBIC imaging, in order to simulate P_{\max} conditions, is tricky and beyond the scope for this work.

[3] On page 6, the authors mainly discussed the EBIC result with an emphasis on the beam-damage-related decay of two different samples (Figure 2). They attributed the differences in EBIC signals to the formation of iodide vacancy, which limited the charge recombination lifetime. The conclusion sounds reasonable, but why the diffusion lengths derived from EBIC showed no significant difference in these samples?

Our reply: We thank the reviewer for this input. We've discussed the relation between beam-induced damage and diffusion length estimates in previous reports, but for completeness sake we have added the following lines to the text (page 7) of the present ms.:

Analysis of beam-induced changes in plan-view EBIC mapping shows degradation of the signal with repetitive exposure. As the plan view imaging does not include any pre-processing of the sample the degradation can be attributed to defect formation due to the electron beam. As previously seen in similar cases, the beam-induced damage does not alter the diffusion length estimation in the first few exposures, but merely lowers the total EBIC signal [ref 13 and doi: 10.1021/acs.jpcllett.5b00889].

Figure 2a shows the comparison of normalized EBIC intensities, but how about the absolute intensities?

Our reply: Absolute intensities are an extremely complicated issue in EBIC measurements as the intensity depends on many parameters, many of which are not related to the sample's properties, such as homogeneity of the cleaved surface. Ultimately, our home-built EBIC setup is not capable of directly outputting currents from the EBIC measurements, but rather only relative intensity (ranging from 1-255 in gray scale). We added a note regarding this issue in our experimental description, page 14:

For each diffusion length measurement (planar view, which does not involve sample cleaving), repetitive scans were performed in order to study the signal decay and compare it to cross-sectional decay as described in the text. Since our setup does not provide absolute EBIC current, we consider relative intensity only.

Are there any differences at grain boundaries and grain internals of the control and w-Pbl2 samples?

Our reply: Further to the answer to the previous comment, grain (domain) boundaries are more prone to morphology-related changes in the signal than to any other suspected changes. We have encountered this phenomenon multiple times and, indeed, we have on several occasions expressed our concern at meetings that too much interpretation depends on (or, too little attention is given to) this very strong artifact. Therefore, we limit our discussion to what are in our experience more solid data, even if this limits the conclusions we can draw.

We note this issue in the revised text, page 9:

We note that the above discussion is limited to bulk grains and the interface between the HaP layer with the selective contacts, and does not include HaP grain boundaries. We also selected only areas of the sample that had a flat morphology. This self-imposed limitation is due to the EBIC signal sensitivity to sample surface morphology.

The authors should have more discussion on the EBIC results to show the contrast between these samples.

Our reply: The following discussion was added (page 6-7):

... The similar excited carrier diffusion lengths in the two samples are not surprising for two main reasons:

1. Both sample types show similar short circuit current (EBIC is done at short-circuit condition, figure S2)
2. The diffusion lengths are upper limits, i.e., for unipolar transport as the second carrier is collected immediately (the lower limit is the layer thickness). Under operating conditions, closer to P_{\max} , a more pronounced difference in diffusion lengths may be seen. Bias application during EBIC imaging, in order to simulate P_{\max} conditions, is tricky and beyond the scope for this work.

... Analysis of beam-induced changes in plan-view EBIC mapping shows degradation of the signal with repetitive exposure. As the plan view imaging does not include any pre-processing of the sample the degradation can be attributed to defect formation due to the electron beam. As previously seen in similar cases, the beam-induced damage does not alter the diffusion length estimation in the first few exposures, but merely lowers the total EBIC signal [ref 13 and doi: 10.1021/acs.jpcclett.5b00889].

On pages 7-8,

... Two possible mechanisms may result in the signal peaking after a few exposures:

(1) Initial passivation of (surface) trap states, which leads to suppression of (near) surface recombination, and which initially increase the EBIC signal. A peak is formed if the beam-induced damage also creates damage to the material, which at some point becomes significant and results in measurable signal reduction;

(2) Beam-induced change in the effective doping concentration, which is initially optimized with respect to the first scan, but later forms an excess, which results in an increased recombination rate and, as a result, a lower EBIC signal.

The latter explanation is independent of whether or not a new surface was created as part of sample preparation for EBIC analysis (i.e., sample cleaving). This means that a peak in EBIC signal with repetitive scans should have appeared both in plan-view as well as in cross section imaging. The appearance of that such a peak in the cross-sectioned samples is only relevant for the first explanation, where the HaP has exposed free surface.

... In light of our understanding of the mechanism for beam-induced changes of the EBIC signal, we conclude that the w/-PbI₂-HaP is less susceptible to defect formation during cleaving and lab air exposure than the control sample.

[4] Figure 2b needs a scale bar to show the size of the SEM/EBIC images.

Our reply: A scale bar was added.

[5] Page 6, Line 152. The authors claimed that junction of perovskite/mp-TiO₂ is an n/n+ type heterojunction. Is there any experimental evidence that perovskite is n-type? What is the estimated carrier concentration for the perovskite layer? Are there any differences for the control and w-PbI₂ perovskite?

Our reply: We thank the reviewer for pointing out this “over-statement” of ours, i.e., using the concept of n/n+ type heterojunction as a fact. While this may well be the case, we should have been more careful and, thus, have corrected the text (page 9) to state our interpretation of the data without invoking an n/n+ heterojunction:

‘An advantage of having the HaP@mp-TiO₂ layer could be to promote charge separation, even before carrier injection into the TiO₂ by formation of a high electron concentration at the HaP/HaP@mp-TiO₂ interface.’

[6] It would be more readable if the major GIWAXS features in Figure 3 that are discussed in the manuscript can be labeled.

Our reply: We modified the figure caption in Figure 3 by describing major GIWAXS features. Diffraction patterns shown in GIWAXS were fully indexed and assigned in Figure S4.

[7] Page 7, Line 168-169. The absence of δ -HaP in the conventional XRD could be due to the formation of δ -HaP on the surface rather than the grain boundaries between PbI₂ and α -HaP.

Our reply: We thank the reviewer for this comment. We revised the sentence by adding the possibility that δ -HaP may be on the surface.

Reviewer #3

Park et al report how the excess of PbI₂ in the perovskite precursors improves the photovoltaic performance of perovskite solar cells. EBIC measurements were applied to estimate the carrier diffusion length and reveal local electrical properties in the devices. The authors also conducted GIWAXS and TEM characterizations to understand how the local crystallite orientation and microscopic morphology correlate with the local electrical properties as revealed by EBIC. The authors found that the excess of PbI₂, which resides in the perovskite grains rather than at grain boundaries, results in low electronic defect density in perovskite films. The excess of PbI₂ also improves the orientation of perovskite crystallites, resulting in better charge collection and suppressed J-V hysteresis. The origin of the excess of PbI₂ in determining solar cell performance has been reported by multiple groups (Nano Lett. 14, 4158-4163(2014); J. Am. Chem. Soc. 138, 10331-10343(2016); Adv. Energy Mater. 6, 1502206(2016); APL Mater. 2, 091101(2014)). The point-of-views as presented by authors in this work, are not quite convincing to me at the current stage, and not supported well by sufficient experimental evidence. Generally speaking, this work is lack of novelty as compared to the reported similar works as mentioned above. Therefore, this paper should be rejected. My detailed comments are listed below.

While the reviewer states

The origin of the excess of PbI₂ in determining solar cell performance has been reported by multiple groups (Nano Lett. 14, 4158-4163(2014); J. Am. Chem. Soc. 138, 10331-10343(2016); Adv. Energy Mater.6, 1502206(2016); APL Mater. 2, 091101(2014)).

Those studies all report on and describe the phenomenon. While some tried to find intrinsic material causes for it, indeed, the “origin” (or “origins”...), that issue is, though, still wide open as otherwise our study would not have been necessary. Moreover, references reviewer mentioned above did not consider the existence of δ -FAPbI₃ and different crystal orientation in relation to excess PbI₂.

Therefore, we respectfully disagree with this opinion.

The reviewer then writes

The point-of-views as presented by authors in this work, are not quite convincing to me at the current stage, and not supported well by sufficient experimental evidence.

Which is certainly fair criticism and we will relate to it in great detail here and in the revised version of the ms.

We respectfully, in view of the above, disagree with the reviewer's subsequent statement

Generally speaking, this work is lack of novelty as compared to the reported similar works as mentioned above.

and, thus, submit that the conclusion

Therefore, this paper should be rejected

should be re-considered.

We now refer to each of the further comments of the reviewer and detail how we have revised the ms., accordingly.

1. The use of EBIC to determine local electrical properties is tricky (especially for unstable hybrid perovskites) due to the electron beam induced damage during measurements (These hybrid perovskites are vulnerable to electron beam irradiation), and significant surface recombinations as a result of damage during the sample preparation.

Our reply: The reviewer is correct, but as this reviewer likely knows, this is preaching to the converted because from the very beginning of this field and use of e-beam-based methods (esp. SEM, CL and EBIC, we have been pointing out exactly what the reviewer writes AND have taken the measures needed, to deal with this problem. This should be eminently clear from our earlier research publications, as well as an account we wrote (Ref 17 in the manuscript). To be able to do these and other e-beam experiments and not measure artefacts (which have indeed appeared in published reports, but... not ours) we took and take steps to get a sense of the effect of these artefacts on the measured properties. In the above-mentioned *Acc. Chem. Res.* (Ref 17) and another paper of ours on comparing MAPbBr₃ to CsPbBr₃, *J. Phys. Chem. Lett.*, 7, 167–172, 2016) and also here, the beam-induced changes can, in some well-defined cases, such as when re-scanning the same image area induces changes in EBIC intensity, be used as a tool to investigate materials properties.

(actually, the authors did not mention how to prepare the cross-section samples in the main text or experimental section).

Our reply: We have added now to the experimental section how the cross-sections were made (page 14):

‘Cross-sectional imaging was performed at 3 kV beam energy, after the sample was mechanically cleaved and transferred to the vacuum chamber within less than 2 minutes. In order to cleave the sample, a small, straight cut is made on the glass (uncovered) side of the substrate across the substrate using a glass-cutting pen. Then, compressive pressure is applied at one of the contacts (FTO or the etched glass/gold) to fracture the device.

2. The variation of the EBIC signal during scans (as shown in Figure 2) is also questionable for the measurements of unstable perovskite materials. This can be reflected by the decrease in EBIC signal after multiple scans, which is probably due to the beam-induced damage during imaging.

Our reply: This is not questionable, as **that is very much the exact point we make.** The control sample is more prone to both sample preparation damage and beam-induced damage. We suggest that this is the result of stabilization of the chemical potential during crystallization by the excess PbI_2 , as discussed on page 8 of the main text.

We added the following to the text, page 8:

‘In light of our understanding of the mechanism for beam-induced changes of the EBIC signal, we conclude that the w/- PbI_2 -HaP is less susceptible to defect formation during cleaving and lab air exposure than the control sample.’

3. For the caption of Figure 2, these sentences may be placed in the wrong place: "Here, the position of guest molecules is unconfirmed. Finally, (III) the intermediate phase film is converted to perovskite phase with corner-sharing octahedrons via extraction of DMSO guest molecules by annealing process. The perovskite film is extremely uniform and flat because of solid-state conversion from the uniform and flat intermediate phase film."

Our reply: We thank the reviewer for drawing our attention to this issue, which is now fixed: the legend is now re-written and most of the confusing text was removed.

The legend now is:

Figure 2. a, Normalized EBIC signal vs. scan number for control-HaP (blue) and w/- PbI_2 -HaP (green). The EBIC signal of each sample is normalized to the maximum intensity of that sample. **b,** Cross-sectional SEM images of the two samples with EBIC mapping of the cross section after various beam exposure.’

4. To justify the reliability of the carrier diffusion length, the authors should present the detailed fitting method and mechanism for the estimation of carrier diffusion length ($L_n = 7.5 \pm 0.2 \text{ um}$ and $L_p = 3.5 \pm 0.8 \text{ um}$)

Our reply: The reviewer is correct and a discussion regarding the fitting was added to the text, pages 6:

We can extract the diffusion length from the EBIC data by fitting the EBICurrent to a simple exponential decay,

$$I = I_0 + Ae^{[-X/L_{n/p}]} \quad (1)$$

where I is the signal intensity, I_0 is the intensity at maximal collection efficiency, A is an arbitrary intensity coefficient, X is a spatial coordinate and $L_{n/p}$ is the diffusion length for electrons/holes, as described by Donolato et al. [<https://doi.org/10.1063/1.343932>]. The 1-D model, on which equation (1) is based, assumes that excited carrier motion under neutral field conditions is limited by the diffusion length. In such a model, where the carrier motion is perpendicular to the line across which carrier separation occurs, the current can be modeled by an exponential decay as described by equation (1).'

5. As shown in the EBIC images of Figure 1a-b, it appears that two layers can be seen in Au side (close to the Au edge, clearly different EBIC contrast in the top as compared to the other part), What is that layer? In addition, several dark spots are observed in the EBIC images (e.g. Figure 1d), what are those dark spots?

Our reply: As noted in the figure caption, the image is plan-view, not a cross-section. The apparent bright layer is an artifact caused when the e-beam scans in parallel to where a large EBIC contrast shift occurs. This bright "layer" disappears when we zoom in and an image is taken, where the contrast shift is more gradual, or if the scan is shifted by 90 deg. The description of the image caption has been altered as follows, to make the figure clearer:

Original:

Figure 1. a, Plan-view images of a device using secondary electron (SE) contrast (left) and electron-beam-induced current (EBIC) measurements (right). **b,** The edges of the Au pad and FTO are indicated for clarity. **c,** Expanded SE (left) and EBIC (right) images at the Au pad edge of the device. The Au pad edge is difficult to resolve in the SE image, but it is clearly exposed by EBIC contrast. The EBIC line profile, averaged from all the pixel lines between the blue lines, is overlaid on the image. **d,** Expanded SE (left) and EBIC (right) images at the FTO edge of the device. Owing to coverage by multiple layers, the FTO edge cannot be resolved in the SE image. Using EBIC contrast, the FTO edge can be located clearly. The EBIC line profile, averaged from all the profiles between the blue lines, is overlaid on the image. To obtain carrier diffusion lengths from these profiles, three positions over two samples were averaged.

Revised:

Figure 1. Plan-view images of a device, using secondary electron (SE) contrast (left) and EBIC measurements (right). In all images, the TiO_2 /perovskite/PTAA (poly-triarylamine) layers are present. The changes are only in the presence or absence of the electrodes to the selective contacts, FTO or Au. **(a)** The edges of the Au pad and FTO are indicated for

clarity. The apparent brighter layer close to the Au edge is an artifact of the scan parallel to an interface where charge extraction occurs. **(b)** Expanded SE (left) and EBIC (right) images at the Au pad edge of the device, which is rotated by 90 degrees with respect to (a). The Au pad edge is difficult to resolve in the SE image, but it is clearly exposed by EBIC contrast. The yellow EBIC line profile, averaged from all the pixel lines between the blue lines, is overlaid on the image. The red line marks the area in the image where the EBIC line profile starts from and collection efficiency is maximal. The derived diffusion length for holes from the EBIC line profile is $3.5 \pm 0.8 \mu\text{m}$. **(c)** Expanded SE (left) and EBIC (right) images at the FTO edge of the device. Owing to coverage by multiple layers, the FTO edge cannot be resolved in the SE image. Using EBIC contrast, the FTO edge can be located clearly. The EBIC line profile, averaged from all the profiles between the blue lines, is overlaid on the image. The derived diffusion length for electrons from the EBIC line profile is $7.5 \pm 0.2 \mu\text{m}$. The apparent dark spots are probably due to discontinuous HaP film at the edges of the FTO. At the bottom left, paths of electron beam-generated charge carriers are schematically illustrated. When the HaP is excited in an area far away from one of the electrodes, the respective carrier has to diffuse laterally to be collected (red arrows; shown as occurring through the HaP, but may also occur through relevant (hole or electron) charge transport layer). The electron beam scans from above and has to penetrate the Au and PTAA layers.

6. For the EBIC images, in order to help to understand the EBIC images, the authors are suggested to provide the intensity scale of EBIC signals.

Our reply: Absolute intensities are an extremely complicated issue in EBIC measurements as the intensity depends on many parameters, many of which are not related to the sample's properties, such as homogeneity of the surface. Our home-built EBIC setup, as many others, does not directly output currents from the EBIC measurements, but only relative intensity (ranging from 1-255 in gray scale). We added a note regarding this issue in our experimental description, page 14:

For each diffusion length measurement (planar view, which does not involve sample cleaving), repetitive scans were performed to study the signal decay and compare it to cross sectional decay as described in the text. Since our setup does not provide absolute EBIC current, we consider relative intensity only. '

7. The diffraction peaks should be indexed for all the GIWAXS patterns (Figure 3).

Our reply: We modified the figure caption in Figure 3 by describing major features of GIWAXS. Diffraction patterns shown in GIWAXS were fully indexed and assigned in Figure S4.

8. The authors claim that "both HaP samples contain three types of crystals, viz. trigonal Pbl₂, hexagonal (δ -), and cubic (α -) HaP". The diffraction peak, should be identified respectively in the GIWAXS pattern (Figure 3).

Our reply: We identified diffraction patterns for trigonal PbI_2 , hexagonal (δ -), and cubic (α -) HaP shown in GIWAXS. Please see the revised figure caption and GIWAXS in Figure 3)

9. *The authors also claim that "We note that in conventional XRD measurements δ -HaP was not observed (Figure S6), which suggests that it may form as very tiny boundary layer between PbI_2 and α -HaP crystals.²⁰ This hypothesis is consistent with the stronger intensities for δ -HaP [100]h in w/- PbI_2 -HaP (Figure 3f) than in control-HaP (Figure 3c)." Which peak is for δ -HaP [100]h ?*

Our reply: We indicated δ -HaP [100]h in the modified GIWAXS.

10. *The key information of how the excess of PbI_2 induces the preferred orientation of the perovskite crystallites is still not clear in this work. And it is not clear how to correlate the orientation of the crystallites with the EBIC in the local region.*

Our reply: We appreciate this helpful comment. In fact, in order to answer these questions, we compared the crystallographic evolution of as-prepared layers deposited with and without excess PbI_2 on substrate using in-situ GIWAXS. As we explained in our manuscript, the first phase which appears during the annealing process is δ -FAPbI₃ but not PbI_2 . The δ -FAPbI₃ has a great influence on the subsequent formation of the HaP crystal, but it is not clear how it affects the crystal orientation. Therefore, the questions by this reviewer, including the correlation with EBIC, should be our next research topics.

11. *The authors claim that the excess PbI_2 resides in the perovskite grains rather than at grain boundaries based on the TEM studies. However, this claim should be revisited. To my understanding, the PbI_2 inclusions are in the perovskite matrix, in the other words, the PbI_2 inclusions are surrounded by the perovskite crystals (or grains). These independent small grains may form a large grain with grain boundaries. Thus, the PbI_2 may be considered as residing in the grain boundaries.*

Our reply: We thank the reviewer for this comment. The reviewer's explanation on the process of PbI_2 inclusion in the HaP matrix is very interesting. However, the explanation may be true if PbI_2 precipitates earlier than HaP during the annealing process of the complex film consisting of the solvent molecule(DMSO)- PbI_2 -FAI (MABr), or if excess PbI_2 is remains unreacted from the beginning. We monitored the process of producing HaP from the complex film by in-situ GIWAXS and found that the δ -HaP phase was first generated and then α -HaP and PbI_2 were produced. This result suggests that the organic cation-deficient complex layer due to excess PbI_2 appears on the δ -HaP with removal of the solvent molecules contained in the complex and is divided into stoichiometric α -HaP and pure PbI_2 during annealing at high temperature. Also, if the initially generated or unreacted PbI_2 is surrounded by perovskite grains, the PbI_2 crystal should be randomly oriented on the GIWAXS 2D pattern. In order to reduce any confusion about PbI_2 generation, we slightly modified the manuscript. Our TEM supports this result.

This finding suggests that the organic cation-deficient complex layer, due to excess PbI_2 , appears on the δ -HaP with removal of the solvent molecules, contained in the complex, and separates into stoichiometric α -HaP and pure PbI_2 during annealing at high temperature.

12. The solar cell device fabrication details should be provided in the experimental section. In addition, the characterization method for J-V curves (photovoltaic performance) should be provided. Since the short-circuit current density is so high, the authors should provide the EQE curves for two devices.

Our reply: We agree with these comments and supplemented the experimental procedure as follows and EQE data in figure S2.

Fabrication of solar cell device. Device Fabrication: A 60 nm thick dense blocking layer of TiO_2 (bl- TiO_2) was deposited onto an F-doped SnO_2 (FTO, Pilkington, TEC8) substrate by spray pyrolysis, using a 20×10^{-3} M titanium diisopropoxide bis(acetylacetonate) solution (Aldrich) at 500 °C, to prevent direct contact between FTO and the hole-conducting layer. 80 nm thick mesoporous TiO_2 (home-made TiO_2 nanoparticles: average particle size \approx 50 nm, crystalline phase = anatase) films were spin-coated onto the bl- TiO_2 /FTO substrate and were calcined at 500 °C for 1 h in air to remove the organic portion. To fabricate efficient perovskite cells based on the $(\text{FAPbI}_3)_{0.85}(\text{MAPbBr}_3)_{0.15}:\text{PbI}_2$, 1.05 M solutions with the ratio of 0.85($\text{NH}_2\text{CH}=\text{NH}_2(=\text{FA})\text{PbI}_3$) and 0.15($\text{CH}_3\text{NH}_3(=\text{MA})\text{PbBr}_3$) (in case of excess PbI_2 7.5 mol% PbI_2 powder was added to this precursor solution) in N-N-dimethylformamide(=DMF) and dimethylsulfoxide(=DMSO) = (6:1 v/v) were then coated onto the mp- TiO_2 /bl- TiO_2 /FTO substrate, heated to 50 °C, by two consecutive spin-coating steps, at 1,000 and 5,000 rpm for 5 and 10 s, respectively. During the second spin-coating step, 1 mL ethyl ether was poured onto the substrate after 5 s. A polytriarylamine (PTAA) (EM index Co. LTD., $M_n = 17500 \text{ g mol}^{-1}$) / toluene (10 mg/1 mL) solution with 7.5 μL Li-bis(trifluoromethanesulfonyl) imide (Li-TFSI) / acetonitrile (170 mg/1 mL) and 4 μL 4-tert-butylpyridine (TBP) added, was spin-coated on the perovskite layer / mp- TiO_2 / bl- TiO_2 / FTO substrate at 3000 rpm for 30 s. Finally, 80 nm of Au was deposited as electrical contact by thermal evaporation. The active area of this electrode was fixed at 0.16 cm^2 .

Characterization of solar cells. The J-V curves were measured with a source meter (Keithley 2420) using a solar simulator (Newport, Oriel Class AAA, 94043A) at 100 $\text{mA}\cdot\text{cm}^{-2}$ illumination (AM 1.5 G) and a calibrated Si reference cell (Newport, Model 91150V) certificated by NREL. The calibrated 1 sun illumination is automatically maintained by exposure control (Newport, Model 68951). The J-V curves were measured in reverse scan modes. The step voltage and delay time were fixed at 10 mV and 40 ms, respectively. The delay time is used at each voltage step before measuring the current. The J-V curves of each device were measured by masking the active area with a metal mask (area of 0.096 cm^2). External quantum efficiencies for each HaP solar cell were measured, using an IQE- 200B system (Newport, Oriel).

Reviewer #4 Review report

General statement

The authors target a relevant topic (PbI₂ in mixed perovskites), which has been shown to be of importance for high efficiency perovskite solar cells, and is something which due to measurement difficulties is something that still not is properly understood.

We fully agree with this assessment of the field, which is in sharp contrast to that of reviewer 2.

They use some advanced characterisation techniques to add more data that can be useful to further understand the importance of PbI₂ in perovskite films. The topic, the scope, and the techniques used makes it of potential interest for nature communications and its readers. I think the work in principle is publishable. There is, however, also problems with the paper and I think it requires a mayor revision before it potentially could be published. The paper is not very well written, the descriptions are a bit confusing, and the results, their implications, and how they are extracted requires better descriptions. Below follows some comments I have that I think should be addressed.

Comments.

The EBIC measurements.

I do not have personal experience with EBIC, but most of the intended readers probably do not have that either. I think the description of the experiments are rather confusing. I would like a better description of things like:

- EBIC can probe this by using this effect*
- With this sample geometry we therefor expect to see this and higher/lower signals for this/that regions*
- If we do, we can based on this reasoning extract this information in this way*
- In our samples we see this*
- Because we see this, we extract this information in this way*
- This information then implies those interesting things*

1. The logic and the flow of the EBIC section must be written in a clearer, and more logical coherent way.

Our reply: We thank the reviewer for this comment. We have re-written the text, added many descriptions to make it sufficiently clear to the non-expert. Due to the length of the script, and the fact that we added discussions in different places, we give below the logic flow, with corresponding pages:

1. EBIC measurements can probe carrier diffusion lengths, L_n and L_p (page 4-5), which is a critical material/device property for solar cells. Here we had to use a special form of EBIC, *plan-view*, on full devices to obtain L_n and L_p , because the devices have a width of the active absorber that is less than the diffusion lengths. As this is not the usual way EBIC is used to extract diffusion lengths, we then:
2. describe how, in these samples, we can use plan view EBIC and give the working assumptions for use of *plan-view* EBIC due to the inability to measure diffusion lengths in cross-sectional EBIC images (page 5).
3. After describing the results, expected from *plan-view* EBIC of a full device, (page 5), we present our experimental EBIC results (page 6).
4. To correlate the signal to L_n and L_p values, we describe how to interpret and fit the data according to a model (page 6) and present the extracted results (page 6).
5. This section ends with a discussion on the EBIC results in our sample geometry (page 6-7).
6. Following the previous discussion, we then show how defect properties can be probed using *cross-sectional* EBIC (page 7), similar to what we did in an earlier report (ref. 17).
7. We then present our *cross-sectional* EBIC results (page 7), followed by mechanisms to interpret the data and discussion on the proposed mechanisms (page 7-8).
8. Lastly, we introduce and discuss differences in EBIC signal at the HaP / mp-TiO₂ interfaces, between the different samples (page 8-9).
9. The final discussion on differences in EBIC between the samples is on page 9.

A slightly abbreviated version of the above appears now on page 3-4 of the revised text.

2. *In figure 1. Is (b) exactly the same as (a) but with annotation?*

Our reply: Yes, we have now combined (b) and (a) together for clarity.

3. *What is the geometry of the sample? Is this a top view or a cross section?*

Our reply: It is a top view of the whole device, as shown in Figure 1 and written in the text on page 3.

4. *You have indicated the Au edge and the FTO edge. Pleas also indicate which part that is Au and which part that is FTO.*

Our reply: Changes have been made to the image to illustrate this.

5. *Is the bottom right Au, and is the bottom left FTO?*

Our reply: Correct; Changes have been made to the image to illustrate this.

6. *If that is so, how is the sample prepared?*

Our reply: Following the previous question, we hope now the image is clearer.

7. *Is the gold lying on of the hole conductor?*

Our reply: Yes, it is a complete device. We've added a statement in the main text (page 4) as well as a discussion regarding this setup on page 4-5:

'... EBIC collection efficiency mapping of a PV device, in this case, fluorine-doped tin oxide (FTO)\dense-TiO₂\mp-TiO₂\Perovskite\PTAA(poly-triarylamine)\Au '

'...In this case an uneven concentration of excited holes and electrons in an already low-doped semiconductor results in a low recombination rate. In order to estimate excited carrier diffusion lengths from plan-view EBIC, we assume significant lateral transport in the PTAA is unlikely due to its high resistance compared to that of the HaP film. This is due to a 2 - 3 orders of magnitude higher hole mobility in the HaP than in the PTAA and also the difference in the layer thickness (>500 nm thick HaP layer and <50 nm for the PTAA).¹¹⁻¹³ Lateral transport of electrons in the d-TiO₂ and HaP@\mp-TiO₂ may be more efficient than that of holes in the PTAA and contribute to the EBIC signal as the e⁻-beam scan away from the FTO (over parts from which the FTO was etched). Reduced concentration of one type of carrier due to fast extraction of one of the carrier types along with point excitation would increase the effective apparent diffusion length of the other carrier by lowering the probability of second order recombination. A cleaner estimate of the exact diffusion lengths in the HaP would require deposition of the HaP layer, on glass, with two selective contacts, which do not sandwich the HaP film. Deposition over glass requires adjusting the HaP deposition process and may result in a different electronic quality material. The estimated upper limit represents the potential of the material under optimal conditions and in a real device.'

8. *If so, how did you manage to not deposit the hole conductor on top of the exposed perovskite?*

Our reply: The hole conductor is found on top of the perovskite. This is now detailed in Figure 1.

9. *If the bottom left is FTO, how did you remove both the perovskite and the TiO₂ under layer?*

Our reply: We did not remove the perovskite or the TiO₂. It is now clearly stated in the figure's legend:

' In all images, the TiO₂\perovskite\PTAA (poly-triarylamine) layers are present. The changes are only in the presence or absence of FTO or Au.'

10. *Figure S.2 give some guidance, but more descriptions is needed to clearly show the reader what was done. Figure S2, or some extended version of it, should definitely be a part of the main manuscript, and could be included in figure 1.*

Our reply: We agree with the reviewer, and a thorough discussion regarding Figure S2 was added to the main text, page 5-6 as detailed in point (7). Furthermore, Figure 1 was altered and a schematic from Figure S2 was added to the figure.

11. Line 74 and 80. You first claim that EBIC require information of the number of defects and then you claim that EBIC indicate that there is less defects. That may be true but logically those two statements does not match when they are presented together.

Our reply: We thank the reviewer for this comment. We apologize for our mistake and removed the first remark regarding defects.

12. Line 95. If the carrier motion is parallel to the built in field, is it then not by definition effected by the electric field? Again, I think this would be clearer with a better description of sample geometry, and the intention behind it.

Our reply: We again thank the comment for this remark. As shown now in Fig. 1 left bottom, the carrier motion is perpendicular (!) to the built in field. The text was corrected.

13. Line 104. Extraction of L_n and L_p . That is a key result and must be described better. You do not show the fit (as you say you do in figure 1). Describe some theory to why it is reasonable to believe that the exponential fit would give those values. This deserved to get some more attention. If you have found any other estimates of the diffusion lengths in the literature, please relate to those.

Our reply: As mentioned in the text, the numbers for L_n and L_p are upper limit estimates for the measured devices. The way to extract L_n and L_p is now described in the text (page 6):

' We can extract the diffusion length from the EBIC data by fitting the EBICurrent to a simple exponential decay,

$$I = I_0 + Ae^{[-X/L_{n/p}]} \quad (1)$$

where I is the signal intensity, I_0 is the intensity at maximal collection efficiency, A is an arbitrary intensity coefficient, X is a spatial coordinate and $L_{n/p}$ is the diffusion length for electrons/holes, as described by Donolato et al.¹⁴ The 1-D model, on which equation (1) is based, assumes that excited carrier motion under neutral field conditions is limited by the diffusion length. In such a model, where the carrier motion is perpendicular to the line across which carrier separation occurs, the current can be modeled by an exponential decay as described by equation (1). Such analysis results in values of $L_n = 7.5 \pm 0.2 \mu\text{m}$ and $L_p = 3.5 \pm 0.8 \mu\text{m}$, as shown in **Figure 1b and c.**

GIWAXS

Line 77-78. Textured in an azimuthal angel orientation in-between 0-90°. To me, this says absolutely nothing at all. If a cubic phase is rotated 0-90° that would cover every possible orientation and there would thus not be an alinement? Does not azimuthal orientation require a fixed point to be referred to make sense? Why use spherical coordinates do describe the orientation of crystals in a thin film? Have you considered using Cartesian coordinates instead? I think that would be clearer.

Our reply: We thank the reviewer for very constructive comment and agree with this comment. As we know, 2D-GIWAXS is a good way to analyze crystal orientation because lateral and normal ordering can be probed at the same time. However, as mentioned by the reviewer, because the data obtained in hemispherical form appears in 2D, the cubic phase does not show any difference in rotation in X, Y, and Z directions. Nevertheless, we fully indexed diffraction patterns shown in 2D-GIWAXS (see Figure S4). Therefore, we believe that our 2D-GIWAXS results clearly describe the orientation of HaPs that depend on excess PbI_2 .

When you talk about orientation of crystals, for it to be interesting you must relate the orientation of the crystals towards the substrate. An illustration would greatly help here and I suggest an additional figure.

Our reply: We supplemented the illustration in figure S7 (c) as shown below.

Concerning the in-situ GIWAXS measurements

Interesting approach. Hopefully more can be done with this in alter studies (increasing number of temperatures to measure, change the heating ramp, comparing different compositions)

I think it is unclear which composition that is measured. With or without excess PbI_2

Our reply: We thank this reviewer very much for the constructive comments. Unfortunately, as you know, the use of GIWAXS, a large facility, is very limited and difficult to measure with many experimental variables. Next time, we will try what the reviewer commented. We monitored in-situ GIWAXS with excess PbI_2 .

TEM

Half of all the TEM images has terrible quality. That is probably a result of problem while converting to pdf during file uploading, but is something that must be fixed.

Our reply: The resolution of the TEM images shown in the original manuscript is very good. The resolution of the figure seems to be low in the process of converting it into pdf.

The domain indicating crystal orientation in the TEM images are about an order of magnitude smaller than the crystal grains as they appear in the SEM images. Can you comment on this?

Our reply: Our results indicate that the crystals that appear as one grain in the SEM are not complete single crystals. The results of TEM observing various domains inside one grain have been reported in recent papers. (references; doi:10.1038/ncomms14547, doi.org/10.1002/adma.201705230)

Perovskites are known to be fragile and delicate materials, and FIB is a rather brutal technique. It would be reasonable to have some discussion concerning the risk for sample damage, and which reasons there are to believe that the prepared samples are representative for the real film.

Our reply: In general, MAPbI_3 system is very difficult to perform FIB sampling for TEM specimens due to the quick damage by high energy ion beam. However, FAPbI_3 and their mixed halide perovskites are somewhat stable during FIB sampling. We also performed FIB sampling by a highly skilled person

Other

Line 40. Also MABr , FABr

Our reply: We revised them in main manuscript.

Line 43-44 (and 22 and 85). The mixed perovskite is not a mixture of two perovskite phases as the used notations indicate. Data show that it is a one-phase compound. It is thus not $[\text{FAPbI}_3]_x[\text{MAPbBr}_3]_{1-x}$ but $\text{FAxMA}_{1-x}\text{PbBr}_3$

Our reply: We partially agree with this comment. It is true that $[\text{FAPbI}_3]_x[\text{MAPbBr}_3]_{1-x}$ becomes single phase, but in some cases, it is reported that phase separation occurs in Br or I rich phase. So, please understand that we are marking our system as $[\text{FAPbI}_3]_x[\text{MAPbBr}_3]_{1-x}$.

Synthesis

There is a reference to a synthesis procedure. I do not think that is good enough. Digital space is cheap, and how the materials were synthesised, and how samples were prepared, are such crucial aspects of the work that they should be described in some detail in the manuscript (even if they are almost identical to previous procedures)

Our reply: Based on this comment, we supplemented the experimental method in detail.

Reviewers' Comments:

Reviewer #2:

Remarks to the Author:

The authors have made substantial revisions which successfully addressed all comments. I recommend publication.

Reviewer #3:

Remarks to the Author:

In this manuscript, the authors report how PbI₂ excess in halide perovskite precursors result in improved solar cell performance. Although this topic is somewhat interesting and important, the results in this work can not answer this question (how PbI₂ excess in halide perovskite precursors result in improved solar cell performance ?) clearly. First, from the film microstructure side, it is unclear how to relate different crystal orientation of HaP with excess PbI₂, and it is also not clear how the different crystal orientation impacts charge extraction. Second, from the device photovoltaic parameters side (e.g. Voc, FF, JSC), it remains unclear how PbI₂ excess in halide perovskite precursors improves solar cell performance. This manuscript is publishable but its importance and novelty do not guarantee its publication in the high-level Nature Communications.

Line 132-150: Authors say the carrier diffusion length in between w/ PbI₂-HaP and control samples is similar. Why are they similar? If the HaP film texturing is different as shown by the authors, the carrier diffusion length could be possibly different. The authors mention that the carrier diffusion length is 7.5 μ m and 3.5 μ m for electron and hole respectively. What are these values for, w/ PbI₂-HaP ? or control sample? Would you please provide the L_n and L_p values for w/ PbI₂-HaP and control sample, respectively? It seems this question has been asked by Reviewer #2, but I did not see the response.

The authors should show the fit to get carrier diffusion length, as also requested by Reviewers #3 and #4.

Why is the w/-PbI₂-HaP less susceptible to defect formation during cleaving and lab air exposure than the control sample ? Is there any direct evidence ?

Is the result of stabilization of the chemical potential during crystallization by the excess PbI₂ ? Is there any evidence ?

Authors mention that in-situ GIWAXS found that the δ -HaP phase was first generated and then α -HaP and PbI₂ were produced. Would you please present a complete in-situ GIWAXS data to show such information ?

Authors say the use of plan-view EBIC is due to the inability to measure diffusion lengths in cross-sectional EBIC images. However, it seems that the authors used the cross-sectional EBIC image to measure diffusion length in authors' previous publication (Ref. 16 Edri, E., Kirmayer, S., Mukhopadhyay, S., Gartsman, K., Hodes, G. & Cahen, D. Elucidating the Charge Carrier Separation and Working Mechanism of CH₃NH₃PbI₃-xCl_x Perovskite Solar Cells. Nat. Commun. 5, 3461 (2014).) Do you have a comment ?

In Figure 2a, the authors claim that a peaking was observed for both control-HaP and w/-PbI₂-HaP samples ? For the control sample, it looks right, but it is not convincing for the case of w/-PbI₂-HaP sample, because only one data point is present before the peaking and the difference between the initial value and the maximum value is very small. It is not quite convincing to say there is increase in EBIC signal during the first few scans for the w/-PbI₂-HaP sample. The authors may need more solid data to draw such conclusion. As performing repetitive scans over the same area,

what is the time interval between two scans ? Is there a time-related variation of EBIC signal ?

The authors claim an HaP content > 25% can clearly be seen in the mp-TiO₂ layer using high resolution Z contrast SEM imaging. How can authors identify the content of HaP in mp-TiO₂ by a cross-sectional SEM imaging ? I guess the authors made such estimation based on the atomic number contrast image. But, if the dark area is for TiO₂, the content of HaP is far more than 25% in the mixture layer.

In Figure 3, where is peak F δ [002]hexagonal ?

The authors mention that HaP crystal transformations from the as-prepared film were monitored from 6 s to 480 s on a hot plate during heating from 30 °C to 150 °C by in-situ GIWAXS, to reveal when PbI₂ remnant was produced, and when the crystalline HaP was oriented. Would you please show the in situ GIWAXS data that reveal when PbI₂ remnant was produced, and when the crystalline HaP was oriented.

What do those red dash lines represent in Figure 4a,d ?

The authors mention that (Line 298-301) optimization of excess PbI₂ in the deposition solution can improve fill factor (FF) and short-circuit current density (JSC) (also, external quantum efficiency), and increase shunt resistance (Rsh). However, it appears that the optimization of excess PbI₂ also improves open circuit voltage (VOC). Would you please comment on why the VOC was improved by excess PbI₂ ? Can authors comment on how PbI₂ excess in halide perovskite precursors result in improved solar cell performance more specifically ? Which photovoltaic parameter is affected most by PbI₂ excess and why ? I guess this is the key information that this paper wants to deliver as emphasized in the manuscript title.

A statistics of device performance between between w/ PbI₂-HaP and control samples should be provided. This is a general agreement in perovskite photovoltaic society especially when comparing the device performance between two types of devices.

Would you please comment on why a w/-PbI₂-HaP precursor solution yields films that fill the mp-TiO₂ better than what is the case for films made with the control solution ? ? Is this because of surface potential of PbI₂ or viscosity of the solution ?

Reviewer #4:

Remarks to the Author:

I have now read the revised manuscript.

I think the authors have done a good job at responding to the questions and comments raised by the referees. I am now in favour of publication.

Reviewer #2

The authors have made substantial revisions which successfully addressed all comments. I recommend publication.

Our reply: We appreciate the excellent comments of this reviewer.

Reviewer #3

In this manuscript, the authors report how PbI₂ excess in halide perovskite precursors result in improved solar cell performance. Although this topic is somewhat interesting and important, the results in this work cannot answer this question (how PbI₂ excess in halide perovskite precursors result in improved solar cell performance ?)_clearly. First, from the film microstructure side, it is unclear how to relate different crystal orientation of HaP with excess PbI₂, and it is also not clear how the different crystal orientation impacts charge extraction. Second, from the device photovoltaic parameters side (e.g. Voc, FF, JSC), it remains unclear how PbI₂ excess in halide perovskite precursors improves solar cell performance. This manuscript is publishable but its importance and novelty do not guarantee its publication in the high-level Nature Communications

We respectfully disagree with the reviewer and, hope that, as the other reviewers before, also this reviewer will be convinced, based on the revisions that we made and the answers that we provide to the questions raised, that are outlined below.

Line 132-150: Authors say the carrier diffusion length in between w/ PbI₂-HaP and control samples is similar. Why are they similar? If the HaP film texturing is different as shown by the authors, the carrier diffusion length could be possibly different.

Our reply: We agree with the reviewer that they could well have been, but we did experiments and do not find differences between the two types of cells (text is now on lines 148-9). We note, though, that, as the diffusion lengths are much larger than the film thickness, they are not limiting factors for performance in either cell.

Thus, there is no inconsistency between different performances and similar diffusion lengths for the two cells.

We discuss this in some depth in the ms. on lines 149-157:

“ The similar excited carrier diffusion lengths in the two samples are not surprising for two main reasons:

1. Both sample types show similar short circuit current (Figure S3, EBIC is done at short-circuit condition.)

2. The diffusion lengths are upper limits, i.e., for unipolar transport as the second carrier is collected immediately (the lower limit is the layer thickness). Under operating conditions, closer to P_{\max} , a more pronounced difference in diffusion lengths may be

seen. Bias application during EBIC imaging, in order to simulate P_{\max} conditions, is tricky and beyond the scope for this work.”

The authors mention that the carrier diffusion length is 7.5 μm and 3.5 μm for electron and hole respectively. What are these values for, w/ $\text{PbI}_2\text{-HaP}$? or control sample? Would you please provide the L_n and L_p values for w/ $\text{PbI}_2\text{-HaP}$ and control sample, respectively? It seems this question has been asked by Reviewer #2, but I did not see the response.

Our reply: We compared and discussed the statistics of L_n and L_p . The values are now given in the Table that is part of the **new Fig. S2**.

Our response in the original revision was:

We added to the experimental (page 14) the following lines:

“Statistics for all EBIC data were obtained from two sets of w/ $\text{PbI}_2\text{-HaP}$ and control samples, prepared independently. Each set was composed of two samples from each type. For diffusion length estimation each sample was measured five times, at different locations. At each location the average of 25 EBIC lines was used to arrive at a single calculated value. The five values from each sample were compared between all samples of the same type, i.e. 20 values per sample type, to form an average value and standard deviation. For the total intensity decay analysis, an averaged EBIC signal was taken from a 2D square map. The map area was 5 consecutive pixel lines across the HaP layer. The intensity decay analysis was performed by repetitive scans of the same area. Next, the intensity at each location was normalized, according to the maximum achieved intensity for that area. An averaged initial intensity value was obtained by a similar averaging process as described above.”

Furthermore, we added the following lines on pages 6-7 in order to prevent any confusion regarding the results of L_n and L_p in Figure 1b and c:

“The similar excited carrier diffusion lengths in the two samples are not surprising for two main reasons:

1. Both sample types show similar short circuit current (EBIC is done at short-circuit condition, figure S11)
2. The diffusion lengths are upper limits, i.e., for unipolar transport as the second carrier is collected immediately (the lower limit is the layer thickness). Under operating conditions, closer to P_{\max} , a more pronounced difference in diffusion lengths may be seen. Bias application during EBIC imaging, in order to simulate P_{\max} conditions, is tricky and beyond the scope for this work.”

The authors should show the fit to get carrier diffusion length, as also requested by Reviewers #3 and #4.

Our reply: As mentioned at the experimental section, 5 line scans are taken at each sample in order to find an average L_n and L_p value. The fitting formula for the data as well as the model it is based on were also shown in the revised ms, page 6 lines 129-138:

“We can extract the diffusion length from the EBIC data by fitting the EBICurrent to a simple exponential decay,

$$I = I_0 + Ae^{[-X/L_{n/p}]} \quad (1)$$

where I is the signal intensity, I_0 is the intensity at maximal collection efficiency, A is an arbitrary intensity coefficient, X is a spatial coordinate and $L_{n/p}$ is the diffusion length for electrons/holes, as described by Donolato et al.¹⁴ The 1-D model, on which equation (1) is based, assumes that excited carrier motion under neutral field conditions is limited by the diffusion length. In such a model, where the carrier motion is perpendicular to the line across which carrier separation occurs, the current can be modeled by an exponential decay as described by equation (1).”

The fitting of the data results in an R^2 value of 0.99 and 0.94 for L_n and L_p . The better fit for L_n is probably the result of the stronger EBIC signal since the e-beam is not attenuated by the Au pad.

We have added the R^2 values to the experimental:

“The fitting of the data results in an R^2 value of 0.99 and 0.94 for L_n and L_p .”

Furthermore, we have added an example to explain the fitting process as well as a table summarizing the fits, to the SI, as Fig. S2.

We have added to the main text, page 6 line 169 the following addition:

“The fitting process is illustrated in Figure S2.”

Why is the w/-PbI₂-HaP less susceptible to defect formation during cleaving and lab air exposure than the control sample ? Is there any direct evidence?

Our reply: We sincerely thank the reviewer for valuable comments. As we know, PbI₂ is not fully ionized into Pb²⁺ and 2I⁻ if dissolved in DMF/DMSO solvent. On the other hand, it has been reported that PbI₂ exists as a so-called iodoplumbate complex that interacts with fully ionized organic cations and halide anions, and its structure or behavior varies with the ratio of PbI₂ to organic-ammonium halide as perovskite precursor. In addition, the initial thin film coated with the perovskite precursor solution forms a quite stable intermediate phase such as (organic cation)₂Pb₃I₈·2DMSO, which can be thermally decomposed and ultimately converted to perovskite crystal thin film, as reported by J. Cao et al. (J. Am. Chem. Soc., 2016, 138 (31),9919–9926). It is therefore readily understood that the ratio of PbI₂ to organic-cation halide in precursor solution will have a significant impact on the final perovskite crystal orientation and defect concentration via the intermediate phase. In other words, the intermediate thin film coated with a solution containing excess PbI₂ relative to the organic cation halide is expected to differ from that coated with the precursor solution dissolved in the stoichiometric ratio in terms of perovskite crystallization and defect concentration. For instance, (organic cation)₂Pb₃I₈·2DMSO intermediate was reported to be an I-deficient phase (ref:J. Am. Chem. Soc., 2016, 138 (31),9919–9926). We have already explained in the manuscript that the crystallization rate and the texturing of the perovskite crystal phase, depending on the presence or absence of excess PbI₂ in the precursor solution. In

this connection, although there is no direct evidence, it is believed that the PbI_2 contained in the crystallized perovskite serves to protect the perovskite phase from environmental change such as cleaving and lab air exposure.

Authors mention that in-situ GIWAXS found that the δ -HaP phase was first generated and then α -HaP and PbI_2 were produced. Would you please present a complete in-situ GIWAXS data to show such information?

Our reply: We replaced a complete *in-situ* GIWAXS data as seen in below.

Authors say the use of plan-view EBIC is due to the inability to measure diffusion lengths in cross-sectional EBIC images. However, it seems that the authors used the cross-sectional EBIC image to measure diffusion length in authors' previous publication (Ref. 16 Edri, E., Kirmayer, S., Mukhopadhyay, S., Gartsman, K., Hodes, G. & Cahen, D. Elucidating the Charge Carrier Separation and Working Mechanism of $\text{CH}_3\text{NH}_3\text{PbI}_3\text{-xCl}_x$ Perovskite Solar Cells. Nat. Commun. 5, 3461 (2014).) Do you have a comment ?

Our reply: We thank the reviewer for giving us the opportunity to clarify further the need we had to develop a novel EBIC approach for these cells.

The devices in ref. 16 were deliberately fabricated to be 2 microns thick to allow us to measure the EBIC signal decays to extract the diffusion lengths in cross-sectional EBIC measurements. In the present report, real high efficiency solar cells were used and for cells to be high-efficiency ones, puts limitson the thickness of the absorber, in this case to 600 nm; we thus see a high and constant EBIC signal between the selective constants on cross sections, allowing only to derive lower limits of diffusion lengths by this method, as noted by us in the original revised m.s, page 6 line 147:

“...The diffusion lengths are upper limits, i.e., for unipolar transport as the second carrier is collected immediately (the lower limit is the layer thickness).”

In Figure 2a, the authors claim that a peaking was observed for both control-HaP and w/-PbI2-HaP samples ? For the control sample, it looks right, but it is not convincing for the case of w/-PbI2-HaP sample, because only one data point is present before the peaking and the difference between the initial value and the maximum value is very small. It is not quite convincing to say there is increase in EBIC signal during the first few scans for the w/-PbI2-HaP sample. The authors may need more solid data to draw such conclusion.

Our reply: We agree with the reviewer that detached from the text of the ms. our conclusion may seem not so convincing. This, however is a result of our selection of the extreme case of signal strength of ~90% of the maximum, to illustrate our point. The text, though, describes a situation where the initial signal of the w/-PbI2-HaP sample is 70% ± 8% (1 stdev = 8%) of the maximum and that the maximum can occur after 3 consecutive scans and 30% is significant and not small.

As performing repetitive scans over the same area, what is the time interval between two scans ?

Our reply: The time between consecutive scans is ~3 minutes (a result of the scan speed and image resolution), and the time held at each image pixel is ~200 µsec.

Is there a time-related variation of EBIC signal?

Our reply: For repetitive scans the sample is held longer and longer under the beam. There is no quantitative, structured work that we are aware of, but we as well as others, have observed that fast scans leave less damage than slow scans. A quick scan is often used to center a region of interest without leaving significant damage. We have published a more extensive review on beam damage previously, which is cited at the ms. (Acc Chem Res Kedem et al paper, ref 17).

If the reviewer meant a dependence on the time between scans, then no beam damage dependence on time between scans was found by us (unpublished). We note that strong

dependence could indicate significant self-healing properties for the material, which we would have been extremely happy to be able to report. Although such properties were observed by us for HaPs, for light-induced damage, and the possibility is very exciting, we do not have at this point the data to support such hypothesis for e-beam exposure. Naturally the time between scans in both sample types is identical.

We now added the following line to the experimental:

“Every image scan takes 2.8-3.0 minutes while re-taking the images is done in less than 3 seconds, in which the current image is recorded and immediately followed by a re-scan of the area”

The authors claim an HaP content > 25% can clearly be seen in the mp-TiO₂ layer using high resolution Z contrast SEM imaging. How can authors identify the content of HaP in mp-TiO₂ by a cross-sectional SEM imaging? I guess the authors made such estimation based on the atomic number contrast image. But, if the dark area is for TiO₂, the content of HaP is far more than 25% in the mixture layer.

Our reply: We agree with the reviewer that the HaP content in the mesoporous layer could be considerably more than 25 % (which is what we write in the text, as also quoted by the reviewer), but it is clearly much less than the HaP content in the over-layer. As the overlayer is not completely homogeneous in thickness, and the mesoporous layer isn't either, and has on top of that an irregular pore/grain size distribution, we cannot be any more accurate than what we have written.

We have changed the text on page 8 line 188 to emphasize this:

“As an HaP content > 25% (but considerably less than the content in the HaP overlayer) can clearly be seen in the mp-TiO₂ layer using high resolution Z contrast SEM imaging (Figure S1), the low EBIC signal is attributed to a high recombination rate.”

In Figure 3, where is peak F δ [002]hexagonal ?

Our reply: We marked F in Figure 3(f) of the revised manuscript.

The authors mention that HaP crystal transformations from the as-prepared film were monitored from 6 s to 480 s on a hot plate during heating from 30 °C to 150 °C by in-situ GIWAXS, to reveal when PbI_2 remnant was produced, and when the crystalline HaP was oriented. Would you please show the in situ GIWAXS data that reveal when PbI_2 remnant was produced, and when the crystalline HaP was oriented?

Our reply: We thank the reviewer for this comment. We supplemented the additional *in-situ* GIWAXS data. To reflect this, we modified the manuscript slightly as the followings:

At the initial stage after 6 s, it was found that α -FAPbI₃ [200] and δ -FAPbI₃ [100] were formed at an early stage. We see that the phase of the δ -FAPbI₃ [100] is significantly reduced and converted to α -FAPbI₃ and PbI₂ [001] phase after 180 s. This implies that an excess PbI₂ in precursor solution forms a thin intermediate layer of the FAI deficient-"FAI-PbI₂-DMSO" complex rather than PbI₂-DMSO, after spin-coating, followed by the conversion of α -FAPbI₃ [100] and PbI₂ [001] phase from δ -FAPbI₃ [100] with annealing at high temperature. This result also explains why PbI₂ is present inside the α -FAPbI₃ grains, as can be seen later in the TEM.

What do those red dash lines represent in Figure 4a,d ?

Our reply: Thank for this comment. The red dotted lines indicate the grain boundaries of the film to be more clearly visible.

The authors mention that (Line 298-301) optimization of excess PbI_2 in the deposition solution can improve fill factor (FF) and short-circuit current density (J_{sc}) (also, external quantum efficiency), and increase shunt resistance (R_{sh}). However, it appears that the optimization of excess PbI_2 also improves open circuit voltage (V_{oc}). Would you please comment on why the V_{oc} was improved by excess PbI_2 ? Can authors comment on how PbI_2 excess in halide perovskite precursors result in improved solar cell performance more specifically? Which photovoltaic parameter is affected most by PbI_2 excess and why? I guess this is the key information that this paper wants to deliver as emphasized in the manuscript title.

Our response: We thank this reviewer for very helpful comments. We absolutely agree that it is the key information to explain how excess PbI₂ can lead to an increase in V_{oc} , and hence overall efficiency. We believe that the following two factors play an important role in improving V_{oc} and hence overall efficiency. First, it was confirmed that the defect concentration of the perovskite film prepared from the precursor solution containing the excess PbI₂ was reduced by examining the intensity change of the EBIC signal according to the number of electron beam exposures. Second factor is that the film deposited with the solution of excess PbI₂ is textured such that the crystal phase has a specific orientation as compared to the control film. The texturing of such crystals may lead to an array of ferroelectric domains that can increase the voltage of the solar cells. As reported (*Nano letter. 2014, 14, 2584-2590*), the presence of ferroelectric domains will result in

internal junctions that may aid separation of photoexcited electron and hole pairs, and reduction of recombination through segregation of charge carriers. Accordingly, the key information obtained in this study is that excess PbI_2 in precursor solution leads to texture the perovskite crystal phase in a specific direction and reduce the defect concentration.

A statistics of device performance between w/ PbI_2 -HaP and control samples should be provided. This is a general agreement in perovskite photovoltaic society especially when comparing the device performance between two types of devices.

Our response: We provided statistics on device performance between w/ PbI_2 -HaP and control samples and modified the SI

Would you please comment on why a w/- PbI_2 -HaP precursor solution yields films that fill the mp-TiO_2 better than what is the case for films made with the control solution? Is this because of surface potential of PbI_2 or viscosity of the solution?

Our response: We appreciate this comment. As we briefly mentioned in the manuscript, and as can be easily understood, changes in the composition of the precursor solution will change the nature of the solution. Of course, in some cases, the changes are too small to be ignored. However, the perovskite precursor solution, in which PbI_2 is not completely

dissolved in the DMF/DMSO solvent (so-called plumbate complex), may vary greatly in the stoichiometric ratio to the organic cation. In fact, DLS measurements shows that the distribution of the size of the plumbate complex in the solution with an excess PbI_2 shifts to a smaller size (please below figure or Figure S12). As the reviewer commented, the viscosity of the solution may be slightly different, but the major factor is that the properties of the plumbate complex change and facilitates the penetration into the small pores of the electrode. We modified the manuscript slightly with this information.

Control HaP precursor

PbI2 excessed HaP precursor

Reviewer #4 Review report

I have now read the revised manuscript.

I think the authors have done a good job at responding to the questions and comments raised by the referees. I am now in favor of publication.

Our reply: We appreciate the excellent comments of this reviewer.

Reviewers' Comments:

Reviewer #3:

Remarks to the Author:

My concerns have been addressed very well. I would like to recommend the publication of this work.